

# Flow acceleration statistics: a new paradigm for wind-driven loads, towards probabilistic turbine design

Mark Kelly[1]

[1] Department of Wind Energy, Danish Technical University, Risø Lab/Campus, Roskilde 4000, Denmark

*Correspondence to*: MKEL@dtu.dk

**Abstract.** A method is developed to identify load-driving events based on filtered flow accelerations, regardless of the event-generating mechanism or specific temporal signature. Low-pass filtering enables calculation of acceleration statistics per characteristic turbine response time; this circumvents the classic problem of small-scale noise dominating the observed accelerations or extremes, while providing a way to deal with different turbines and controllers. Not only is the flow

acceleration physically meaningful, but its use also removes the need for de-trending. Through consideration of the 99th percentile ($P_{99}$) of filtered acceleration per each 10-minute period, we avoid assumptions about distributions of fluctuations or turbulence, and derive statistics of load-driving accelerations for offshore conditions from 'fast' (10 and 20 Hz) measurements spanning more than 15 years. These statistics depend on low-pass filter frequency (reciprocal of turbine response time), but in a nontrivial manner varying with height due to the influence of the atmospheric boundary-layer's

capping inversion as well as the surface.

We find long-term probability distributions of 10-minute $P_{99}$ of filtered accelerations, which drive loads ranging from fatigue to ultimate; this also includes joint distributions of the $P_{99}$ with 10-minute mean wind speed ($U$) or standard deviation of horizontal wind speed fluctuations ($\sigma_s$). The long-term mean and mode of the $P_{99}$ of streamwise accelerations, conditioned on $\sigma_s$ and $U$, are found to vary monotonically with $\sigma_s$ and $U$ respectively; this corroborates the IEC 61400-1 prescriptions for

fatigue design-load cases. An analogous relationship is also seen between lateral (directional) accelerations and standard deviation of direction, particularly for sub-mesoscale fluctuations.

The largest (extreme) $P_{99}$ of filtered accelerations are seen to be independent of 10-minute mean speeds, and with only limited connection to 10-minute $\sigma_s$; traditional 10-minute statistics cannot be translated into extreme load-driving acceleration statistics. From measurement heights of 100 m and 160 m, timeseries of the 10 most extreme acceleration

events per 1 m s$^{-1}$ wind speed bin were further investigated; events of diverse character were found to arise from numerous mechanisms, ranging from non-turbulent to turbulent flow regimes, also depending on the filter scale. Different behaviors were noted in the lateral and streamwise directions at different heights, though a small fraction of these events exhibited extreme amplitudes for both horizontal acceleration components and/or were observed at both heights within a given 10-minute window. Via fits to the tails of the marginal $P_{99}$ distributions, curves of offshore extreme $P_{99}$ of filtered accelerations

for return periods up to 50 years were calculated, for three characteristic turbine response times (filter scales) at the observation heights of 100 m and 160 m.



To drive aeroelastic simulations, Mann-model parameters were also calculated from the timeseries of the most extreme events, allowing constrained simulations embedding the recorded events. To facilitate this for typical industrial measurements which lack three-dimensional anemometry, a new technique for obtaining Mann-model turbulence parameters

was also created; this was employed to find the parameters corresponding to the background flow behind the identified extremes and their timeseries. Further, a method was created to use the extreme acceleration statistics in stochastic simulations for application to loads, including interpretation within the context of the IEC 61400-1 standard. Preliminary parallel work has documented aeroelastic simulations conducted using the extreme event timeseries identified here, as well as Monte Carlo simulations based on the extreme statistics and new method for stochastic generation of acceleration events.

## 1 Introduction and Background

As set out by the IEC 61400-1 standard (IEC, 2019) for wind turbine design, fatigue load conditions are simulated in common industrial practice via the so-called normal turbulence model ['NTM'], with testing of extreme loads due to transients prescribed via an extreme turbulence model ['ETM'] or using simplified scenarios such as the extreme operating

gust ['EOG'] that are considered representative of critical parts of turbine design-load envelopes. The magnitude of the wind events in 'extreme' scenarios is prescribed by the IEC standard in terms of 10-minute statistics, particularly the mean wind speed[1] $U$ and standard deviation of wind speed $\sigma_s$ or its longitudinal (streamwise) component $\sigma_u$, which are known to drive fatigue loads (Dimitrov $et\ al.$, 2018). However, a growing trend towards improved turbine design has been to associate the statistics of observed phenomena (which can involve the wind as well as the turbine and electric grid) with individual

design-load cases [DLCs] in the 61400-1 standard, and now through the emerging 61400-9 standard for probabilistic design. This has been motivated by limitations in the IEC's prescriptions for extreme cases (e.g. Dimitrov $et\ al.$ 2017; Hannesdóttir $et\ al.$ 2017, 2019) as well as the stochastic nature of extremes and reliability analysis (e.g. van Eijk $et\ al.$ 2017; Nielsen $et\ al.$ 2023).

A basis for statistical characterization efforts and constrained turbulence simulation was given by Nielsen $et\ al.$ (2004),

who examined gust examples and occurrence rates of wind speed jumps, and identified the potential need for filtering in such characterization; however, their analysis was essentially limited to the surface-layer regime (10m heights) where large accelerations are inextricably intertwined with ground-affected turbulence. They also identified some events with nonstationary wind and direction timeseries which they attributed to frontal passages, and which did not appear to give large accelerations compared to the DLC's associated with wind direction changes in the IEC 61400-1 standard. However, the

results were affected by the limited amount of data, and eventually superseded by later work such as that of Hannesdóttir $et$

---

[1] Here we follow the convention of using $U$ to denote mean wind speed, following from a coordinate system defined such that the mean wind defines the $x$-direction with velocity component $u$ and lateral velocity component $v$ in the $y$-direction so that $s = (u^2 + v^2)^{1/2}$, with the capitalization denoting 10-minute mean (thus $V = 0$).





*al.* (2019). Hansen & Larsen (2007) made early comparisons of measurements to the IEC's extreme DLC for coherent gust with direction change [ECD], focusing on the joint occurrence of jumps in wind speed and direction; however, they were limited by the small number of observations of joint events. Larsen & Hansen (2008) offered calibration of several IEC extreme DLC's, but they assumed extreme events to be turbulence-driven and connected with 10-minute statistics, as in the
IEC 61400-1 standard.

Hannesdóttir & Kelly (2019) directly detected wind speed ramp events at heights of contemporary turbines ($z \geq 100$m) with a broad range of rise times and magnitudes, comparing their statistics to the ECD design-load case; they showed that direction changes due to such events may exceed the IEC prescription. Hannesdóttir *et al*. (2019) found these events to not exceed the IEC's extreme turbulence prescription, except for some events crossing rated speed (for a particular turbine and
controller) which gave tower-base fore-aft loads exceeding DLC1.3 of the 61400-1. The ramp amplitudes crossing rated speed appeared to be driving the excessive loads in their aeroelastic simulations of single turbines. That extreme loads from wind ramps crossing rated speed are driven by *accelerations* was specifically confirmed by Kelly *et al.* (2021); first obtaining long-term marginal and joint distributions of ramp (bulk) accelerations, pre-ramp speed, and upper-rotor shear for offshore wind ramp events, they used the joint PDFs to generate a representative ensemble of coupled large-eddy and
aeroelastic simulations[2] for an offshore wind farm. The simulations showed most of the observed wind ramps, whose inferred thicknesses spanned ~500 m to 10 km, to persist through the wind farm; they further showed the largest thrust-based loads to occur during maximal accelerations crossing rated speed.

With the above as motivation — most simply the finding that load-driving forces on turbine blades can arise from flow acceleration ("F equals $m \cdot a$") — we investigate offshore flow accelerations and practically applicable statistics derived
from them, along with connections to typical 10-minute means and standard deviations; this is done for both streamwise and lateral (directional) fluctuations. We note that although a couple of studies aimed at statistics of gust-like events have recently appeared in the literature, they did not focus on offshore load-inducing flow characterization at turbine rotor heights. Shu *et al*. (2021) found statistical distributions for different wind gust characteristics including rise times and amplitudes, but they did not consider the associated accelerations (nor the need for or effect of low-pass filtering), and their observations
were from an onshore site with hills upwind. Cook (2023) also considered the 'big picture' of gust events, reviewing and comparing numerous techniques for identifying and classifying them towards establishing an automated method; he found inclusion of additional variables (temperature, pressure) to improve gust classification for extreme value analysis, and gave insight into removal of anomalous spikes and use of sonic anemometry. However, Cook's (2023) study considered only surface-layer wind speeds onshore, examining maximum wind speeds rather than accelerations. Civil engineering literature
has addressed gusts in the design of offshore structures for decades (EuroCode/CES, 2010; ESDU, 2012), but again this has been only in the surface-layer, assuming gusts follow turbulence statistics, and has not considered flow accelerations (despite estimates of structural accelerations). But our focus is on load-driving accelerations in the flow regimes at typical wind

---

[2] Two model-chains of coupled simulations were used: both started with constrained turbulence simulations, with one coupled to an aeroelastic model, and one driving large-eddy simulations coupled with aeroelastic models.



turbine hub heights offshore (100 m and above); such flows differ extensively compared to near-surface flow, which is dominated by turbulence associated with the surface, even more so over rough ground and terrain onshore. Further, in contrast to wind speed statistics, accelerations literally represent the forcing of the flow on turbine structures; as described later below, they do not require detrending, and with low-pass filtering their statistics are computable per different wind turbine systems and responses.

The structure of the remaining parts of the paper is as follows: Section 2 outlines the data and its use, gives the methodology's basis, and demonstrates the methodology with associated statistical metrics. Section 3 presents results, showing the long-term statistics of the flow accelerations that dominate each 10-minute period, considering both the frequent values which induce fatigue loads and extremes which can be associated with ultimate turbine loads. The extreme flow accelerations are examined further in Sections 3.3–3.4, including their deviation from behavior prescribed in the IEC standard and the different (often non-turbulent) flow regimes associated with them; forms are given for extrapolation of measured statistics to 50-year periods, towards siting and probabilistic turbine design. For practical use, two appendices are offered, connected with Section 3.4. Appendix A gives a method to obtain Mann-model turbulence parameters for the flow behind extreme accelerations events, facilitating constrained simulation of such gust-like events, as well as allowing one to obtain turbulence parameters from typical industrial measurements for general use; Appendix B gives a recipe for synthesis of timeseries with extreme offshore flow accelerations based on the extreme acceleration distributions, including a method for probabilistic operating gusts that accounts for distributions of gust duration and its connection with acceleration amplitude. Section 4 discusses and interprets the findings, with conclusions and implications, as well as ongoing/future work.

## 2 Data and Methodology

As discussed in the introduction above, flow *accelerations* have been found to drive thrust-based loads during ramp-like events in operating conditions. Since wind speed ramps at turbine heights have rise times mostly ranging from roughly 10 s to ~300 s (Kelly *et al.* 2021), and gust durations in particular have been observed to be shorter (below 100 s and most commonly 10–20 s, as in Shu *et al.* 2021[3]), we expect that 10-minute statistics might not be adequate for capturing extreme flow accelerations. 'Fast' data (typically output by anemometers at frequency of ~1 Hz or higher) is known to be needed to capture gusts, as long documented in the civil/wind engineering literature (e.g., Davis & Newstein, 1968), meteorology (e.g., Beljaars, 1987), and recently for windspeed ramps by Hannesdóttir & Kelly (2019). In light of this and because mechanisms other than wind ramps (including phenomena other than turbulence) can cause peak loads on operating turbines, we choose a

---

[3] Note the Shu *et al.* (2021) study was over land, with rise times found at 160 m height. We expect longer rise times offshore, but not by much for such heights beyond the atmospheric surface layer; rise times offshore are not expected to be more than one order of magnitude longer than onshore at such heights, so the argument for 'fast' data still holds.





statistical methodology, instead of attempting to detect or identify events with specific signatures or corresponding to particular physical mechanisms. I.e., we build a statistical characterization of wind accelerations at heights impacting offshore wind turbines, towards universal description. This is done using more than 15 years (Oct. 2004–Mar. 2020) of

high-frequency observations from the Høvsøre turbine test station, located on the west coast of Denmark (Peña *et al.*, 2019). We select data in the offshore flow regime, defining it over the range of wind directions from 240°–300° (which are also the most common for this wind climate) and heights that are unaffected by the coastline, which basically runs in the North-South direction. Primarily one mast was employed, which provided 10 Hz speed (cup-anemometer) and direction (wind vane) data from heights of 100 m and 160 m; the mast's lower sensors (at 60m and 10m) were not used, due to their measurements

being affected by the coastline and land below. A secondary mast 400 m to the south (the same distance to the coastline, with measurements up to 116.5 m) was also used in a supplementary manner, exploiting its three-dimensional sonic anemometers at 80 m and 100 m to test three-component turbulence calculations; these 'sonics' had sample rates of 20 Hz.

As a starting basis for our investigation, we consider load-driving events without joint occurrence of irregular turbine conditions, i.e., away from cut-in and cut-out. Previous studies found the loads induced by ramp accelerations tended to be

largest around rated speed, which tends to be ~11–13 m s$^{-1}$ for multi-megawatt turbines; further, due to the sample rates of 10–20 Hz and the need to calculate many quantities including multiple Fourier transforms for each 10-minute period, we limited the number of samples processed due to computational constraints. Accordingly, to highlight flows where rated speed is crossed and due to the immense amount of data, we select 10-minute periods using the criterion $(8 \text{ m s}^{-1} + \sigma_s) \leq U \leq (18 \text{ m s}^{-1} - \sigma_s)$ where $\sigma_s$ is the standard deviation of horizontal wind speed; the criteria $\sigma_s > 0.3 \text{ m s}^{-1}$ and $\sigma_\varphi >$

$0 \text{ rad s}^{-1}$ were also used to eliminate rare frozen anemometer and wind vane issues, where $\sigma_\varphi$ is the standard deviation of wind direction.

Nielsen *et al.* (2004) and others have noted that to examine statistics of wind speed jumps, one needs to filter the wind timeseries. However, here we note additional details needed to facilitate analysis of load-driving accelerations. First, one must take care with calculating accelerations from timeseries: simple finite-differences do not suffice due to their oscillatory

spectral signature, impacting acceleration statistics in a nontrivial manner (especially the largest per each 10-minute period, which is our main interest). Calculating accelerations directly in Fourier space without approximation avoids this issue:

$$\dot{s} \equiv ds/dt = \mathcal{F}^{-1}[(2\pi f)^2 S_{ss}(f)], \tag{1}$$

where $\mathcal{F}^{-1}$ denotes the inverse Fourier transform, $S_{ss} \equiv |\mathcal{F}[s(t)]|^2$ is the power spectrum of the horizontal speed fluctuations $s$, and $f$ denotes temporal frequency. An example of acceleration spectra, from a 10-minute record including

large accelerations that cross rated speed, is given in Figure 1. The figure displays spectra using different methods to calculate $\dot{s}$ measured by a cup anemometer, starting with the exact calculation in Fourier space (thick blue line). Using a first-order finite-difference $\Delta s/\Delta t$ (dashed magenta line) to approximate $\dot{s}$, one can see significant noise at frequencies above ~0.03 Hz; such noise can lead to spurious peaks in timeseries of the resultant approximate $\dot{s}$, impacting the largest calculated





accelerations. Higher-order finite differences can somewhat improve upon the first-order approximation, but still have
issues; to be simple and exact, we choose *direct* spectral calculation of acceleration timeseries in this study, using (1).

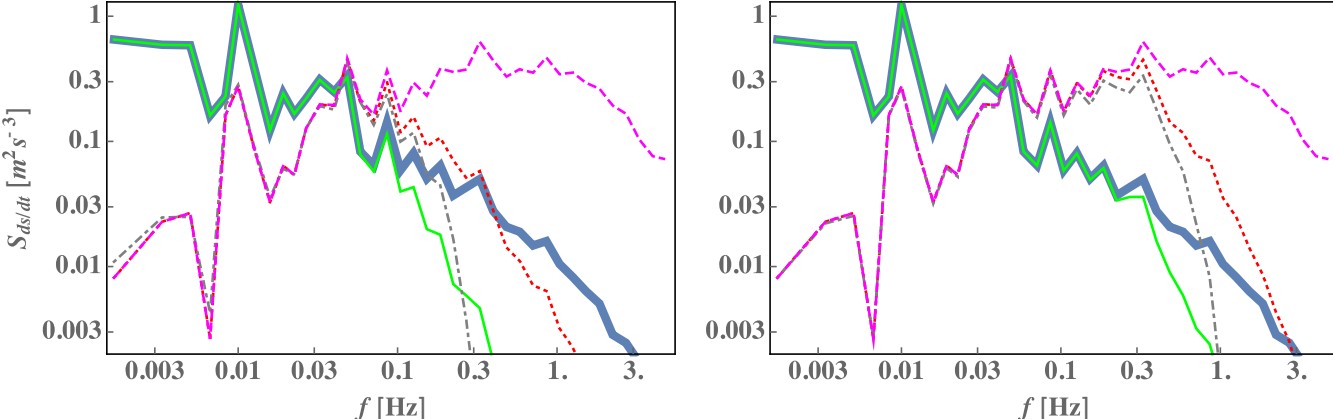

**Figure 1: Spectrum of horizontal flow acceleration from one 10-minute timeseries of cup-anemometer with 10 Hz sample rate, calculated via different methods. Thick blue is $ds/dt$ spectrum $f^2 S_{ss}(f)$; solid green is low-pass filtered $ds/dt$ using 2nd-order Butterworth; dashed magenta is $\Delta s/\Delta t$ via first-order finite-difference; dotted-red is Butterworth O(2) low-pass filtered version of this $\Delta s/\Delta t$; gray dash-dot is digital differentiator with Blackman-Nuttall window. Spectral smoothing of 12 points/decade is done to cleanly display the effects. Left: low-pass filter has $f_c = 1/10$ Hz; right: low-pass filter has $f_c = 1/3$ Hz.**

In addition to calculating accelerations via spectrally based derivative, these need to be appropriately filtered to accommodate the characteristic response of wind turbines, to avoid the small-scale 'noise' that does not impact multi-
megawatt HAWTs due to their size. Figure 1 also displays spectra of low-pass filtered accelerations calculated directly (green) and via finite-difference (dotted red), where a second-order Butterworth filter was used. The left plot shows the filtered acceleration spectra calculated with a filter-frequency $f_c = 1/10$ Hz, while the right-hand graph shows them using $f_c = 1/3$ Hz. One can see that low-pass filtered spectra of the finite-differenced approximation $\Delta s/\Delta t$ also possess significant inflation of fluctuations at moderately small scales ($f > \sim 0.05$ Hz) compared to the unfiltered $\dot{s}$ spectrum, as
well as some suppression of larger-scale fluctuations. Alternately we show the low-pass filtered acceleration spectrum calculated via digital differentiator filter (grey dash-dotted lines) using the same $f_c$; like the finite-difference approximation, it displays spurious addition of noise at moderately high frequencies, albeit with a sharper spectral roll-off.[4] From the figure it is also evident that such strong artificial high-frequency fluctuations are introduced by the finite-difference approximations at high frequencies (particularly for the higher filter frequency $f_c = 1/3$ Hz), that the corresponding low-pass filtered
spectral amplitudes can exceed even the *unfiltered* exact acceleration spectrum; these lead to large false accelerations, which

---

[4] The differentiator filter included a Blackman-Nuttall window, implemented in the software Mathematica; we note that different windowing does not robustly remove the spurious noise at frequencies above ~0.05 Hz. We also point out that although finite-differencing includes an implicit low-pass filter, this is only significant for $f \gg 1/\Delta t$, which here would require weighted averaging over many timesteps; this is more complicated and difficult to control compared to *explicitly* applying a filter to the spectrally-derived acceleration $f^2 S_{ss}(f)$ and taking the inverse-FFT to get timeseries of $ds/dt$.





is another reason that we both recommend and use direct spectral calculation of accelerations henceforth. To allow for different turbine response times, we calculate statistics for three different low-pass filter scales $f_c$, i.e., effective response times $f_c^{-1} = \{30\text{s}, 10\text{s}, 3\text{s}\}$ using a second-order Butterworth filter.[5]

Besides direct spectral calculation of filtered accelerations, for building meaningful statistics of load-driving events we consider the top 1% of accelerations per each 10-minute period. In other words, we calculate $P_{99}$ of the filtered $\dot{s}$ for every 10-minute record; from the collections of such $\dot{s}_{99}$ we can calculate long-term statistics, per different characteristic timescales $f_c^{-1}$. Using $\dot{s}_{99}$ is preferable to 10-minute *maxima* of accelerations, because the latter are less certain (more likely to be outliers). E.g., with a sampling rate of 10 Hz (not unusual for cup anemometers), $\dot{s}_{99}$ corresponds to the $60^{\text{th}}$ largest value; there is considerably less statistical scatter in values which occur (at least) 60 times per 10-minute period, compared to a maximum which occurs just once per period. Alternately one can consider $\dot{s}_{90}$, i.e., the top 10% of accelerations for each 10-minute period. For robustness, in this work we use $\dot{s}_{99}$ as a metric for the flow accelerations expected to drive loads.

It is worth noting that we start by considering statistics of horizontal speed (such as $\dot{s}_{99}$), because the standard instrument used in industrial wind measurement campaigns — the cup anemometer — measures fluctuations and variances in $s$, not the streamwise velocity component $u$. To be more bluntl: although the IEC 61400-1 standard prescribes use of the standard deviation of velocity components, which are dominated by the streamwise one ($\sigma_u$), in industrial practice what is presumed to be $\sigma_u$ is not typically measured as such. To the contrary, from the IEC 61400-50-1, 61400-12-2, and 61400-12-1 standards, measurement of $\sigma_s$ is prescribed when using cup anemometers, as this is what they measure (Kristensen, 2000; Yahaya & Frangi, 2004). To actually obtain $\sigma_u$ requires the use of high-frequency wind vane measurements to find the corresponding high-frequency timeseries $u(t)$, which permits calculation of 10-minute $\sigma_u$; however, the physical separation between anemometer and wind vane can cause a directionally-dependent lag between wind direction and speed, which causes a problem for measuring short-duration events. Addressing this issue is beyond the scope of the current work, and fortunately has negligible impact on extreme events, as we see in Section 3 below.

## 2.1 Preliminary demonstration of statistical methodology

To illustrate the methodology and statistical metric described above, Figure 2 shows how the maximum, $99^{\text{th}}$ percentile, and $90^{\text{th}}$ percentile of low-pass filtered streamwise accelerations vary with reciprocal of filter frequency (characteristic response time), using both second-order and sixth-order Butterworth filters. This is presented for two different 10-minute periods of wind speeds measured at 160m height: a 'typical' period corresponding to the most commonly observed $\dot{s}_{99}$ and $\dot{s}_{90}$ (peak of the long-term distributions of $\dot{s}_{99}$ or $\dot{s}_{90}$), as well as a plot for a 10-minute period corresponding to the one of the strongest

---

[5] We also include some comparisons below involving sixth-order Butterworth filter, but the filter order was not crucial.



streamwise flow-acceleration events[6] within the 16-year dataset. We see from Figure 2 that the filter order has a relatively
small effect on $\dot{s}_{90}$ and $\dot{s}_{99}$ compared to its effect on the 10-minute maximum of filtered acceleration, particularly for filter
timescales of 10 s or shorter; this is further justification for avoiding $\max\{ds/dt\}$ as a metric. It also suggests that getting
representative flow-driving acceleration statistics will be easier using filter scales $f_c$ of 0.1 Hz or 1/3 Hz, in contrast with
choosing 1/30 Hz; the latter corresponds to a characteristic response time of 30 s, which is longer than that corresponding to

commercial wind turbines.

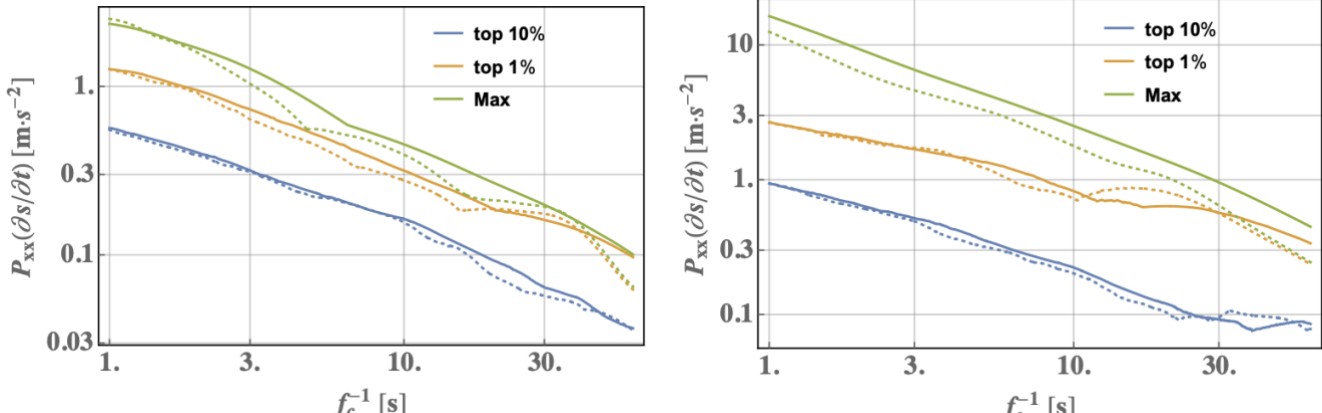

**Figure 2. Dependences of 90th percentile, 99th percentile, and maximum of low-pass filtered acceleration, versus reciprocal of filter frequency (characteristic response time) for wind speeds recorded at 160m height over two different 10-minute periods. Solid and dotted lines indicate 2nd- and 6th-order low-pass filtered accelerations, respectively, using low-pass Butterworth filter.**
**Left: typical/common record; right: case with extreme accelerations (highest in 12–13 m s$^{-1}$ wind speed bin for dataset).**

We note that while Nielsen *et al.* (2004) identified a need for a "3rd order filter to avoid cascading," where the latter refers
to an increasing number of jumps counted with shorter durations, they were concerned with counting *occurrences* of wind
speed *crossing above a threshold* (progressively zooming in to the threshold, at smaller and smaller scales more crossings

are found). Here we are not counting crossing events, but rather calculating exceedance statistics of filtered accelerations for
each 10-minute period; using $\dot{s}_{99}$ (or $\dot{s}_{90}$) avoids the need for a higher-order filter, and as shown above and in Figure 2, the
filter order does not significantly impact this statistic.

---

[6] The case shown in the right-hand plot of Figure 2 corresponds to the 10-minute period containing the largest filtered $ds/dt$
(for filter scales of 1/30, 1/10, and 1/3 Hz) in the mean wind speed bin of 12–13 m s$^{-1}$.



# 3 Analysis and results

Towards probabilistic turbine design, using the filtered flow-acceleration statistics and measurements introduced above we will derive a climatology of (long-term) offshore load-driving accelerations and associated exceedance rates. This includes long-term statistics of filtered $ds/dt$, as well as filtered directional accelerations via the derivative of wind direction, $d\varphi/dt$.

## 3.1 Horizontal and streamwise flow accelerations

Because cup anemometers are the standard instrument used for wind energy in tandem with wind vanes (much more often
than 2- or 3-dimensional sonic anemometers), we continue with temporal derivatives of *horizontal wind speed, $\dot{s}$*; later we will also consider streamwise and lateral components of acceleration (i.e., filtered $du/dt$ and $dv/dt$).

The long-term probability density of 10-minute 99[th]-percentile low-pass filtered horizontal accelerations, $P(\dot{s}_{99})$, is plotted in Figure 3 for the two primary heights considered (100m and 160m above the sea), and for three different low-pass filter scales $f_c$ (1/3, 1/10, and 1/30 Hz). This gives context for the previous figure, showing that the right-hand plot of Figure 2
corresponds to a 10-minute period containing an extreme acceleration event, while the left-hand plot in Figure 2 matches a period where the filtered acceleration is near the most commonly occurring 10-minute filtered $\dot{s}_{99}$; from the two plots in Figure 2, for a filter frequency of 0.1 Hz the $\dot{s}_{99}$ at 160m are approximately 0.3 m s[−2] and 0.7 m s[−2], respectively.

The range of wind speeds considered (8–18 m s[−1]) occur 61% of the time for fully offshore flow at the Høvsøre site, so the actual *number* of $\dot{s}_{99}$ values detected for a given acceleration 'bin' (increment) over the years analyzed would be larger than
detected. Here we assume the *probability density $P(\dot{s}_{99})$* shown in Figure 3 would *not* change if we could include wind speeds below 8 m s[−1] and above 18 m s[−1], i.e., no significant bias arising from differences between $P(\dot{s}_{99}|U < 8 \text{ m s}^{-1})$ and $P(\dot{s}_{99}|U > 18 \text{ m s}^{-1})$. One can further note from Figure 3 that larger $\dot{s}_{99}$ are seen at 100 m compared to 160 m, including the most extreme accelerations. However, because atmospheric boundary-layer [ABL] depths shallower than 200 m are uncommon offshore (Liu & Liang, 2010; Ratnam & Basha, 2010), it is possible that for heights above 160 m, the $\dot{s}_{99}$ could
be yet larger than at 100 m — due to effects from the ABL-capping inversion.

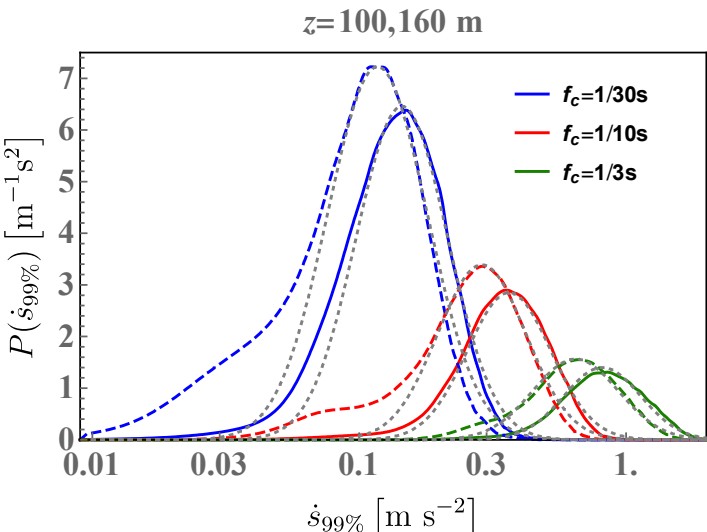

**Figure 3. Long-term distribution of flow accelerations: PDF of 10-minute P99's of filtered d$s$/d$t$, for three different filter scales, from measurements at 100m (solid) and 160m height (dashed). Dotted gray lines are log-normal fits.**


The PDF of $\dot{s}_{99}$ follows a log-normal form around its peak, so the most commonly occurring (fatigue) load-driving accelerations can be approximated with the oft-used log-normal distribution

$$P(\dot{s}_{99}) \;=\; \left(\dot{s}_{99}\sigma_g\sqrt{2\pi}\right)^{-1} \exp\left[\frac{-1}{2}\left(\frac{\ln(\dot{s}_{99})-\mu_g}{\sigma_g}\right)^2\right] . \tag{2}$$

Here the dimensionless geometric (multiplicative) standard deviation parameter is defined through $e^{\sigma_g^2} \equiv 1 + \left(\sigma_{\dot{s}_{99}}/\mu_{\dot{s}_{99}}\right)^2$

and the analogous geometric-mean[7] parameter is $e^{\mu_g} \equiv \mu_{\dot{s}_{99}}e^{-\sigma_g^2/2} = \mu_{\dot{s}_{99}}\left[1 + \left(\sigma_{\dot{s}_{99}}/\mu_{\dot{s}_{99}}\right)^2\right]^{-1/2}$; the latter is equivalent to

the commonly-seen definition $\mu_g \equiv \ln\left[\mu_{\dot{s}_{99}}^2/\left(\mu_{\dot{s}_{99}}^2 + \sigma_{\dot{s}_{99}}^2\right)^{1/2}\right]$. Fits around the peak give log-normal parameter values

$\{\mu_g, \sigma_g\}$ for each filter scale considered, at $z$=100 m and 160 m, which are shown in Table 1. The peaks seen in Figure 3

correspond to the mode of $\dot{s}_{99}$. We note that although analytically $\mathrm{Mo}\{\dot{s}_{99}\} = e^{\mu_g - \sigma_g^2} = \mu_{\dot{s}_{99}}\left[1 + \left(\sigma_{\dot{s}_{99}}/\mu_{\dot{s}_{99}}\right)^2\right]^{-3/2}$ for the

log-normal form, this is not used and the mode is found numerically (via histogram) because the tails of $P(\dot{s}_{99})$ do not follow

the same distribution.

---

[7] Note dimensional consistency in $\mu_g$ can be seen by rewriting $P(\dot{s}_{99}) = \left(\dot{s}_{99}\sigma_g\sqrt{2\pi}\right)^{-1}\exp\left[\frac{-1}{2}\left(\frac{\ln\left(\dot{s}_{99}/\mu_{\dot{s}_{99}}\right)}{\sigma_g} + \frac{\sigma_g}{2}\right)^2\right]$. Our

use of the log-normal form involves fits that do not directly employ $\mu_{\dot{s}_{99}}$, so we use the typical engineering form (2) with $\mu_g$.





**Table 1. Log-normal parameters fitted around peak (most common values) of $\dot{s}_{99}$ distributions**

|  | $\underline{f_c = 1/30\ \text{Hz}}$ | $\underline{f_c = 0.1\ \text{Hz}}$ | $\underline{f_c = 1/3\ \text{Hz}}$ |
|---|---|---|---|
| $\{\mu_g, \sigma_g\}$ at 100 m height: | $\{-1.9, 0.40\}$ | $\{-0.88, 0.36\}$ | $\{-0.1, 0.33\}$ |
| $\{\mu_g, \sigma_g\}$ at 160 m height: | $\{-1.9, 0.32\}$ | $\{-1.1, 0.38\}$ | $\{-0.3, 0.37\}$ |

The largest observed $\dot{s}_{99}$ do not conform to the overall log-normal fits, as the latter capture the most likely $\dot{s}_{99}$ around the
peak of its distributions. However, the extreme tails of $P(\dot{s}_{99})$ are also seen to follow a *different* log-normal distribution,
simply with different log-normal parameter values (for the heights and filter scales considered). The extreme $\dot{s}_{99}$ are shown
in Figure 4, which displays the exceedance probability (i.e., survival function[8] 'SF') of $\dot{s}_{99}$ for the two heights and three filter
scales considered. Fitting to the largest $\dot{s}_{99}$ for all three filter scales at both 100 m and 160 m heights, we obtain the log-
normal parameters $\{\mu_e, \sigma_e\}$ for extreme flow accelerations; their values are shown in Table 2, and the distributions
corresponding to these log-normal fits are shown by grey dotted lines in Figure 4.

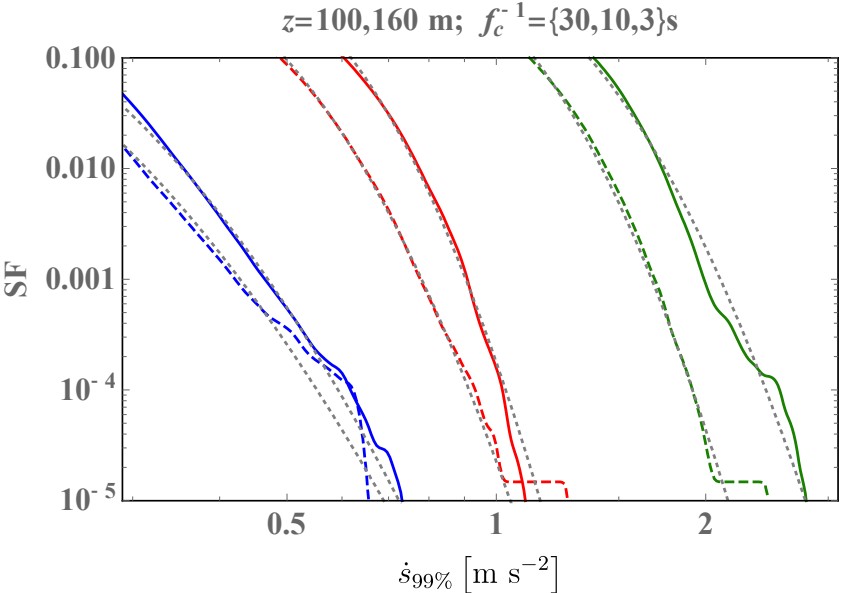

**Figure 4: Survival function ( SF = 1 − CDF) of 10-minute $P_{99}$ of streamwise filtered accelerations (i.e. $\dot{s}_{99\%}$), for the two heights (solid lines are 100 m, dashed is 160 m) and three filter scales ($f_c^{-1}$=30 s in blue, 10 s in red, 3 s in green) considered; gray dotted lines are the extreme fits using a log-normal distribution.**

From the figure one sees that for the top 1–10 values of $\dot{s}_{99}$ the plots can become irregular due the rarity of such events ($<$
once per year), deviating somewhat from the log-normal fits; this is also expected noting that SF = $10^{-5}$ corresponds to one
occurrence in roughly two years, whereas for the range of directions and speeds considered from the 16-year dataset we have
67,648 ten-minute periods or a minimum SF of approximately $1.5 \times 10^{-5}$. We note that, unlike what one might expect from

---

[8] The survival function ("SF") is also equal to 1 minus the cumulative distribution function ("CDF").





inertial-range turbulence, the curves of extreme $P(\dot{s}_{99})$ at both heights cannot be collapsed through a simple relation in terms

of filter scale $f_c$; scaling the 1% most extreme $P(\dot{s}_{99})$ by a factor $f_c^{0.62}$ causes coincidence of only the $f_c$ =1/3 and 1/30 Hz curves at 100 m height, and the $f_c$ =1/3 and 1/10 Hz curves at 160 m height (not shown). We expect this because different mechanisms, with different timescales, are responsible for extreme flow-acceleration events at different heights due to the relative distance to the capping inversion; this is examined later in Section 3.3.2.

**Table 2: Fitted log-normal distribution parameters for extrapolation of highest (extreme) accelerations.**

|  | $f_c = 1/30$ Hz | $f_c = 0.1$ Hz | $f_c = 1/3$ Hz |
|---|---|---|---|
| $\{\mu_e, \sigma_e\}$ at 100 m height: | $\{-1.9, 0.37\}$ | $\{-0.75, 0.21\}$ | $\{0.0, 0.24\}$ |
| $\{\mu_e, \sigma_e\}$ at 160 m height: | $\{-2.1, 0.41\}$ | $\{-1.02, 0.25\}$ | $\{-0.15, 0.22\}$ |

The joint distribution of 10-minute $\dot{s}_{99}$ and mean wind speed, i.e., $P(\dot{s}_{99}, U)$, gives more information about the character of streamwise load-driving accelerations; this shown in Figure 5 for the measurements from 100 m height, for filter scales of 0.1 Hz and 1/3 Hz. We note that essentially the same plots result for 160 m height (not shown), though with slightly smaller

accelerations. The figure indicates that stronger $\dot{s}_{99}$ generally occur for higher wind speeds. The most commonly occurring 10-minute filtered $\dot{s}_{99}$ exhibit an exponential dependence on wind speed: the mode conditional on speed is described by $\mathrm{Mo}\{\dot{s}_{99}|U\} \approx a_{\mathrm{md}}\, e^{-(U/U_{\mathrm{md}})}$ where $U_{\mathrm{md}} \simeq 9$ m s$^{-1}$ and $a_{\mathrm{md}} \simeq 0.3 \cdot \mathrm{Mo}\{\dot{s}_{99}\}$, which for the cases at $z = 100$ m in Figure 5 is $a_{\mathrm{md}}|_{f_c=0.1\mathrm{Hz}} \approx 0.11$ m s$^{-2}$ and $a_{\mathrm{md}}|_{f_c=1/3\mathrm{Hz}} \approx 0.25$ m s$^{-2}$. The 90[th] and 99[th] percentile of $\dot{s}_{99}$ conditioned on $U$ are seen to grow approximately linearly with $U$, while the 99.9[th] percentile of $P(\dot{s}_{99}|U)$ grows more slowly with speed. Nevertheless, no

clear speed dependence for the most extreme $\dot{s}_{99}$ has been found (at either height), as seen in Figure 5. One might assume this to be a sampling artifact and presume extreme $\dot{s}_{99}$ to grow with $U$ like the 99.9[th] percentile of $P(\dot{s}_{99}|U)$; however, as will be seen below in section 3.3, the timeseries for the most extreme $\dot{s}_{99}$ show that the largest flow accelerations can be separate from the 'background' flow or turbulence. Furthermore, the slower growth of the top 0.1% of $P(\dot{s}_{99}|U)$ with $U$ compared to the top 1% of $P(\dot{s}_{99}|U)$ implies that the more extreme accelerations will have a weaker dependence on wind speed, and we

also see in the figure that a larger number of extreme events occur between 14–16 m s$^{-1}$ compared to 12–14 m s$^{-1}$; thus it appears most likely that extreme $\dot{s}_{99}$ are essentially independent of wind speed at these heights, for the range of speeds analyzed.



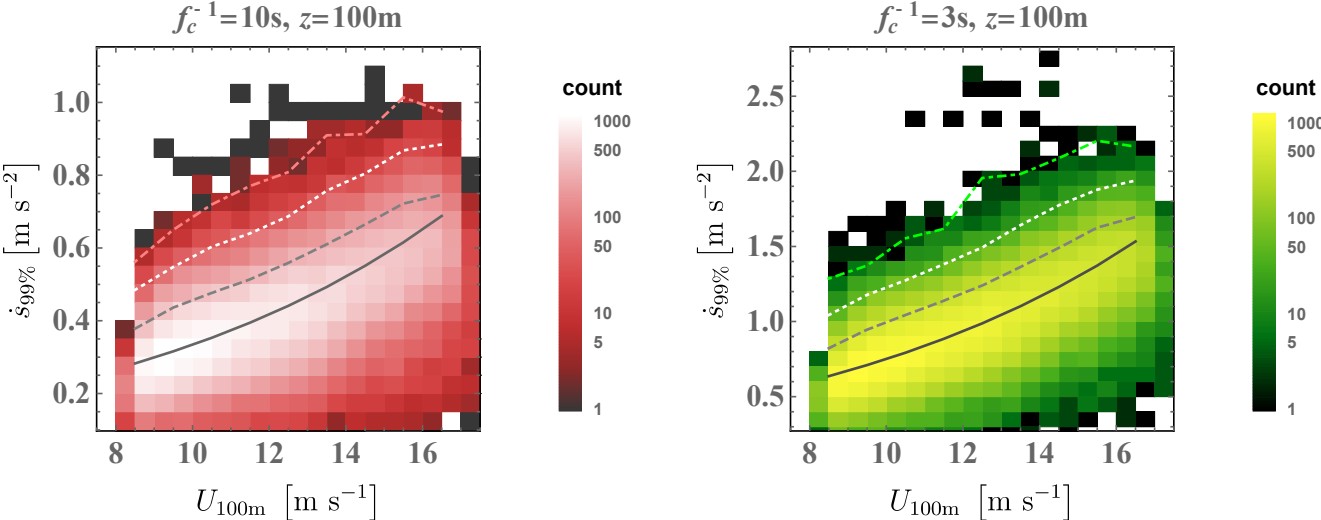

**Figure 5. Joint distribution of 10-minute $P_{99}$ of filtered d$s$/d$t$ and mean wind speed at 100m height, for low-pass filter scales of 0.1 Hz (left) and 1/3 Hz (right). Solid line is exponential fit to the mode of $\dot{s}_{99}$ conditioned on $U$, gray dashed is 90th percentile, white dotted is 99th percentile, and color dash-dotted is 99.9th percentile of $P(\dot{s}_{99}|U)$.**

The long-term behavior of $\dot{s}_{99}$, considering its potential relation to turbine loads, can be further examined in terms of the commonly used 10-minute standard deviation of wind speed $\sigma_s$ (or alternately streamwise velocity $\sigma_u$). Figure 6 shows the joint distribution $P(\dot{s}_{99}, \sigma_s)$ for low-pass filter scales $f_c$ of 0.1 Hz and 1/3 Hz, at both 100 m and 160 m heights. The correlation of all $\{\dot{s}_{99}, \sigma_s\}$ are found to range from 0.73 for $f_c = 1/3$ Hz to 0.83 for $f_c = 1/30$ Hz, with commonly occurring $\dot{s}_{99}$ and $\sigma_s$ exhibiting yet higher correlation; the plots exhibit a simple monotonic dependence of $\dot{s}_{99}$ on $\sigma_s$ around the conditional mode of $\dot{s}_{99}|\sigma_s$. For values of $\sigma_s$ between 0.4 and 1.3 m s (where the jCDF is between roughly 5% and 95%, i.e., rejecting less common joint values) the mode follows a power-law, $\text{Mo}\{\dot{s}_{99}|\sigma_s\} \approx c_\sigma \sigma_s^\beta$, where $\beta$ ranges from 0.75 to 0.8 for the range of $f_c$ and heights analyzed; the constant is $c_\sigma \approx 1.1$ for $f_c = 1/3$ Hz and $c_\sigma \approx 0.4$ for $f_c = 1/10$ Hz. One could try to derive a similar relation based on idealized theoretical arguments, but we avoid such here; this is because we do not know the horizontal turbulence length scale for every 10-minute period, and additionally we would also need to account for the non-Gaussianity which commonly occurs (shown later below). The simple monotonic dependence of frequently occurring $\dot{s}_{99}$ on $\sigma_s$ and their high correlation is expected, since for typical turbulent offshore flow (over a homogeneous surface), larger variability in accelerations reflected by $\dot{s}_{99}$ is connected with larger variability in speed. This result, along with the analogous behavior with wind speed seen in Figure 5, is also consistent with the IEC 61400-1 standard's prescriptions for fatigue loads based on $\sigma_s$ and $\sigma_s \propto U$. However, Figure 6 also shows that more extreme accelerations do *not* appear to exhibit a dependence on $\sigma_s$; although the largest $\dot{s}_{99}$ tend to occur for $\sigma_s$ above ~0.5 m s⁻¹, the maximum observed $\dot{s}_{99}$ are flat over the range of observed $\sigma_s$ from 0.5 to 4 m s⁻¹.



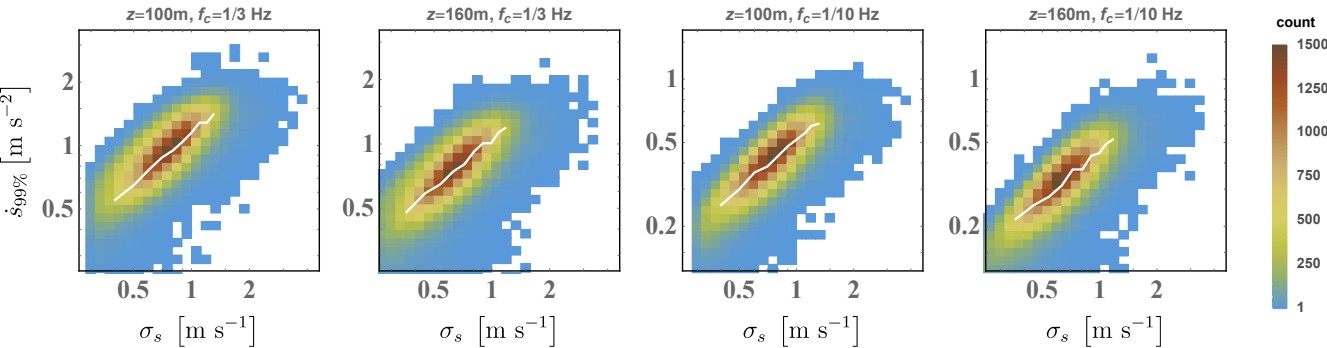

**Figure 6: Joint PDFs $P(\dot{s}_{99}, \sigma_s)$ of all 10-minute $P_{99}$ of filtered acceleration and standard deviation of wind speed, at both heights analyzed, for low-pass filter scales $f_c$ = 0.1 Hz and $f_c$ = 1/3 Hz. White lines show mode of $\dot{s}_{99}$ conditioned on $\sigma_s$.**

As noted by Larsen & Hansen (2014), for fatigue loads it is important to separate microscale wind speed variability from larger-scale (mesoscale) fluctuations; this is conventionally done by removing the latter through 10-minute trend removal[9], which is simply a form of high-pass filtering. Thus we now examine the relationship between $\dot{s}_{99}$ and high-pass filtered $\sigma_s$; using a second-order high-pass Butterworth filter, we remove mesoscale fluctuations in the frequency domain with a high-pass length scale $\ell_{cH} = 2\pi/k_{cH} = 2$ km via Taylor's hypothesis ($f_{cH} = U/\ell_{cH}$) as in Hannesdóttir & Kelly (2019), to get a

"microscale" variance $\sigma_{s,HP}$ for every 10-minute timeseries analyzed. Analogous to the $P(\dot{s}_{99}, \sigma_s)$ shown in Figure 6, the joint distribution of $\dot{s}_{99}$ and $\sigma_{s,HP}$ is displayed in Figure 7. From the figure, we see that $\dot{s}_{99}$ follows $\sigma_{s,HP}$ more closely than it tracks $\sigma_s$. This is expected, since accelerations can be seen spectrally as a sort of high-pass filtered wind speed following Eq. 1 (i.e., multiplication of the spectrum of speeds by $f^2$ removes trends and low-frequency fluctuations). Accordingly, the correlation between $\dot{s}_{99}$ and $\sigma_{s,HP}$ is found to be 0.9±0.02, higher than between $\dot{s}_{99}$ and $\sigma_s$;

counterintuitively, the spatial high-pass filtering also causes the $\{\dot{s}_{99}, \sigma_{s,HP}\}$ correlation to become essentially independent of $f_c$. Furthermore, the conditional mode $\mathrm{Mo}\{\dot{s}_{99}|\sigma_{s,HP}\}$ grows almost linearly with $\sigma_{s,HP}$ (power-law with exponent $\beta$ ranging from 0.9 to 1.03), with the linear approximation $\mathrm{Mo}\{\dot{s}_{99}|\sigma_{s,HP}\} \approx c_{\sigma HP}\sigma_{s,HP}$ having a proportionality constant[10] of order 1. This is again consistent with the IEC 61400-1 prescription to use de-trended wind speed standard deviation. In contrast, from Figure 7 we see that extreme $\dot{s}_{99}$ do not exhibit a clear dependence on $\sigma_{s,HP}$, similar to Figure 6 for

$P(\dot{s}_{99}, \sigma_s)$. However, the same level of extreme accelerations occurs over a narrower range of $\sigma_{s,HP}$ compared to $\sigma_s$. The lack of clear dependence on $\sigma_{s,HP}$ (as well as $\sigma_s$) may be due in part to limited sampling, whereby a larger data set (yet longer measurement period) might indicate (some) increase of extreme $\dot{s}_{99}$ with $\sigma_{s,HP}$ for $\sigma_{s,HP} \gtrsim 1$ m s$^{-1}$.

---

[9] Section 11.3.4 of the IEC 61400-1 standard (2019) suggests wind speed data to be "preferably linearly de-trended".

[10] The linear scaling constant follows $c_{\sigma HP} \approx \left(3.8 \text{ Hz}^{7/4}\right) f_c^{-3/4}$, though there is no theoretical basis for this empirical form.




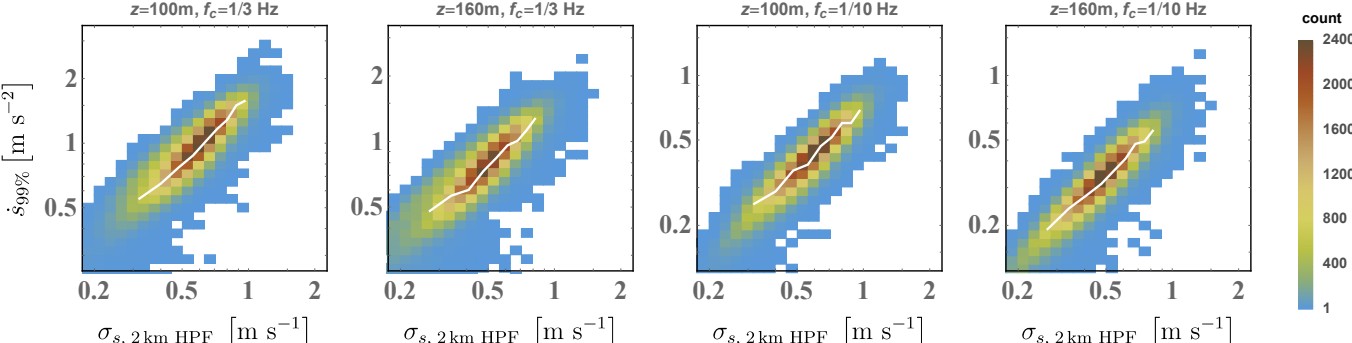

**Figure 7: Joint PDFs $P\left(\dot{s}_{99}, \sigma_{s,3km\ \text{HPF}}\right)$ of all 10-minute $P_{99}$ of filtered acceleration and standard deviation of spatially high-pass filtered wind speed $\sigma_{s,3km\ \text{HPF}}$, where scales larger than 2 km are filtered out of $\sigma_s$. As in Figure 6, results given at both heights analyzed for acceleration low-pass filter scales $f_c = 0.1$ Hz and $f_c = 1/3$ Hz. White lines show mode of $\dot{s}_{99}$ conditioned on $\sigma_{s,3km\ \text{HPF}}$.**

Previous works have presumed that extreme load-driving flow phenomena tend to be associated with non-Gaussian turbulence (e.g., Moriarty *et al.*, 2004; Nielsen *et al.* 2004), and due to non-stationary conditions (Chen, *et al.*, 2007). However, as demonstrated in Figure 8, the observations appear to indicate the opposite, in terms of the skewness of horizontal speed ($\text{Sk}_s$): the most frequently occurring values of $\text{Sk}_s$ show a modestly non-Gaussian behavior, with $\text{Sk}_s < 0$. The figure displays the joint distribution of $\dot{s}_{99}$ for $f_c = 1/3$ Hz with skewness and kurtosis, respectively, at both 100 m and 160 m heights. Overall $\dot{s}_{99}$ does not show a correlation with $\text{Sk}_s$, and extreme $\dot{s}_{99}$ are coincident with vanishing or slightly negative $\text{Sk}_s$, appearing to follow from the same distribution of $\text{Sk}_s$ as more commonly occurring values. Figure 8 also shows the joint distribution of $\dot{s}_{99}$ and kurtosis of wind speed ($\text{Ku}_s$), which indicates that $\dot{s}_{99}$ shows no correlation with $\text{Ku}_s$, and that both the most common and extreme values of 10-minute $P_{99}$ of flow acceleration correspond to the Gaussian kurtosis value $\text{Ku}_s = 3$. We further note that the same qualitative results occur for $\dot{s}_{99}$ calculated using lower $f_c$ (not shown). The lack of correlation between extreme $\dot{s}_{99}$ and either $\text{Sk}_s$ or $\text{Ku}_s$, as well as limited dependence of large $\dot{s}_{99}$ on $\sigma_s$, suggest that a given extreme $\dot{s}_{99}$ is not necessarily connected with the underlying distribution of wind speed fluctuations or turbulence in the associated 10-minute period. Rather, an *event* is caused by one or more flow phenomena that includes a large acceleration with duration much smaller than 10 minutes, which is effectively superposed on the 'background' flow and its fluctuations. This is confirmed directly in section 3.3 below, by examining observed timeseries corresponding to the largest recorded $\dot{s}_{99}$; there we will also see that many extreme events are not associated with non-stationary conditions, i.e., there is not necessarily a jump from some lower speed to a higher one. Further, it follows that (at least some) extreme load-driving accelerations *cannot* be predicted using 10-minute statistics.





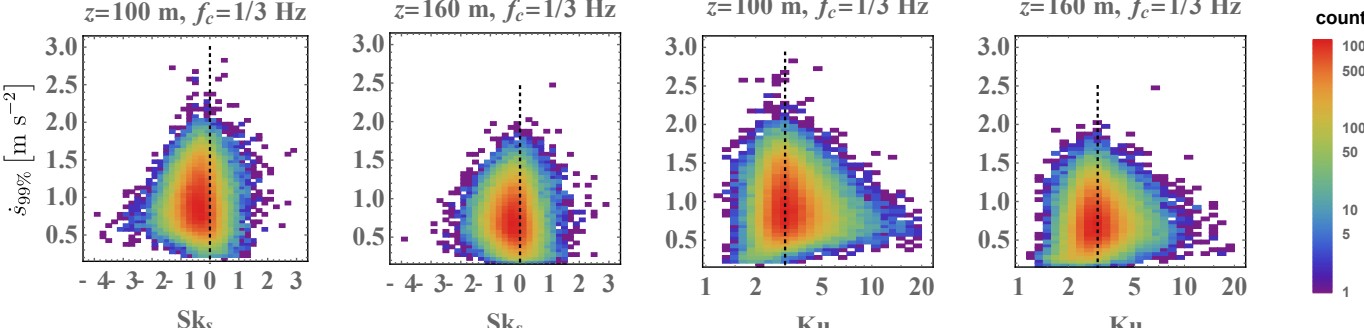

**Figure 8: Joint PDF of $P_{99}$ of low-pass filtered accelerations, with either Skewness or Kurtosis of wind speed; results shown for both 100 m and 160 m heights. Logarithmic scale used for jPDF colors. Dotted lines denote Gaussian values ($Sk_s$=0 and $Ku_s$=3).**


### 3.1.1 Vertical differences

Statistics of the difference in acceleration between 100 m and 160 m were also calculated using the three $f_c$ considered, including the 99[th] percentile of horizontal (and streamwise) accelerations. The most common $P_{99}$ of $\Delta\dot{s}$ (including the mode) were found to be proportional to and ~15–30% higher than $\dot{s}_{99}$ averaged over the two heights, with more variability found

for lower $f_c$. This is shown in Figure 9. The figure also shows that extreme values of $P_{99}(\Delta\dot{s})$ can be twice as large as the $\dot{s}_{99}$ averaged over the two heights. However, such statistics do not account for *when* the respective accelerations occurred at the two heights within each 10-minute period. To do so requires further analysis, due to the time lag associated with the physical shape of acceleration-inducing flow structures (e.g., ramp-like events detected by Hannesdóttir *et al*., 2017, or inclination angles of cold fronts); identifying the signatures of individual events within a 10-minute period and matching these at

multiple heights (e.g. Suomi, *et al*., 2015) or locations is beyond the scope of this work. Still, the differences over a $\Delta z$ of 60 m may lead to large flap-wise blade root bending moments for both fatigue and ultimate loads. As with the $\dot{s}_{99}$ investigated earlier, for $P_{99}(\Delta\dot{s})$ across two heights the high-pass spatially filtered standard deviation of wind speed is a reasonable surrogate for commonly occurring $P_{99}(\Delta\dot{s})$, due to the linear relationship exhibited between $P_{99}(\Delta\dot{s})$ and $\dot{s}_{99}$ around the conditional mode $Mo\{P_{99}(\Delta\dot{s})|\dot{s}_{99}\}$; this is also consistent with the IEC 61400-1 standard's use of detrended $\sigma_s$

for design-load cases that drive flap-wise bending moments.





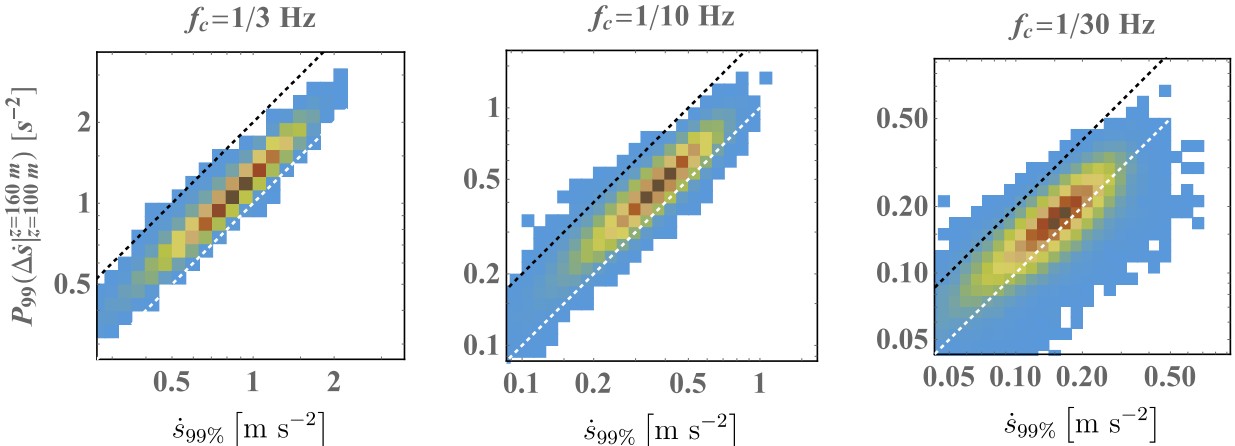

**Figure 9:** $P_{99}$ **of difference in low-pass filtered accelerations between 100 m and 160 m, versus** $P_{99}$ **of low-pass filtered accelerations averaged over the two heights (plotted joint PDF), for all three low-pass filter scales considered. Dotted white line shows 1:1 relationship, dotted black is 2:1.**


Regarding shear, we also report that there is no pattern or correlation between $\Delta U/\Delta z$ and $P_{99}$ of $\Delta \dot{s}$ or $\dot{s}$. This again points toward accelerations that are associated with events separate from the underlying flow or turbulence (when the flow is turbulent), with the acceleration-causing events in effect superposed upon the 'background'.

## 3.2 Directional and transverse flow accelerations

Following the streamwise accelerations considered in the previous sections, here we consider lateral accelerations, expressible through the time rate of change of direction $\dot{\varphi} \equiv d\varphi/dt$. In Cartesian coordinates (denoted by subscript 'c') defined by the mean wind direction at a given height the direction is defined by $\varphi_c \equiv \arctan(v_c/u_c)$, which gives $\dot{v} = \dot{\varphi}_c u[1 + (v/u)^2] + \dot{u}v/u = (\dot{\varphi}_c s^2 + \dot{u}v)/u$; using units of radians per second for $\dot{\varphi}$ then gives $\dot{v}$ in the conventional unit 405 of acceleration, m s$^{-2}$. In the coordinate system used in wind energy (based on incoming wind direction, increasing clockwise) one then has $\dot{\varphi} = -\dot{\varphi}_c$ so that $\dot{v} = (-\dot{\varphi}s^2 + \dot{u}v)/u$. Since the horizontal acceleration $d\left(\sqrt{u^2 + v^2}\right)/dt$ can be written as

$$\dot{s} = (\dot{u}u + \dot{v}v)/s, \tag{3}$$

one can then express the lateral component of acceleration as

$$\dot{v} = (-\dot{\varphi}u + \dot{s}v/s) = (-\dot{\varphi}s\cos\varphi + \dot{s}\sin\varphi) \tag{4}$$

and the streamwise component as

$$\dot{u} = (\dot{s}u/s - \dot{\varphi}v) = (\dot{s}\cos\varphi - \dot{\varphi}s\sin\varphi). \tag{5}$$

Typically, $v/u \ll 1$ so that $\dot{v} \simeq -\dot{\varphi}u \approx -\dot{\varphi}s$, though for strong lateral fluctuations or low wind speeds this approximation might be expected to become inaccurate. However, for the range of wind speeds considered here (8–18 m s$^{-1}$) and noting that





we will be using statistics of the strongest 1% of accelerations from each 10-minute period (denoted $\dot{\varphi}_{99}$), this approximation

becomes reasonable.[11] The strongest 1% of $\dot{\varphi}$ can be either positive or negative, since it is found that the top 1% leftward

and rightward accelerations in a given 10-minute period are on average the same (symmetric), i.e. flipping the sign of $\dot{\varphi}$ we

find its CDF above 0.99 (or SF below 0.01, i.e., the top 1%) is unchanged.

As with $\dot{s}$ analyzed above, the quantity $\dot{\varphi}$ is calculated in the Fourier domain to avoid spurious values that can arise due to

finite-differencing. We again apply a *spatial* 2nd-order Butterworth high-pass filter via Taylor's hypothesis with filter

frequency $f_c = U/2$ km, to decompose the standard deviation $\sigma_\varphi$ into mesoscale and microscale components, where the

Yamartino (1984) method is employed to calculate standard deviations of direction.

Like $\sigma_s$, the 10-minute standard deviation of direction $\sigma_\varphi$ is dictated most often by fluctuations having spatial extents

smaller than 2 km (microscale), with a minority of cases having larger-scale fluctuations dominate. On the other hand, strong

variability in speed or direction at rotor heights (here 100–160 m) tends to be more associated with mesoscale structures —

and not microscale turbulence. This is shown by the left-hand plots in Figure 10, which indicate that for $\sigma_\varphi \gtrsim 10°$ the

mesoscale portion of $\sigma_\varphi$ (larger than 2 km) exceeds the microscale part of $\sigma_\varphi$, and similarly for $\sigma_s \gtrsim 2$ m s$^{-1}$ the mesoscale

part of $\sigma_s$ exceeds the microscale part. However, this is *not* the case for the dominant *accelerations*, as demonstrated by the

right-hand plots of Figure 10, which visualize the statistics of 10-minute $P_{99}$ of temporally low-pass filtered $\dot{s}$ (i.e., $\dot{s}_{99}$, third

plot) and directional acceleration (approximated[12] via $\dot{\varphi}U$, right-most plot). The plots show that there is little correlation

between these $P_{99}$ and $\sigma_{\varphi,\text{meso}}/\sigma_{\varphi,\text{micro}}$, with the extreme accelerations particularly independent of $\sigma_{\varphi,\text{meso}}/\sigma_{\varphi,\text{micro}}$. This is

consistent with extreme flow accelerations having temporal scales longer than ~3 s but shorter than ~2 minutes (via Taylor's

hypothesis, 2 km divided by the highest speeds analyzed 18 m s$^{-1}$). The results shown in Figure 10 are for $f_c = 1/3$ Hz at

100 m height, but the same results occur at 160 m height and for the other $f_c$ (0.1 Hz and 1/30 Hz) for which $\dot{\varphi}_{99}$ and $\dot{s}_{99}$

were calculated.

---

[11] The approximation is found to be valid within 15% for the largest observed lateral acceleration events ($\dot{\varphi}_{99}$) using 3d sonic
anemometer data at 80 m height, and it gives an error smaller than 5% for 325 of the top 360 events (c.f. section 3.3).
[12] Since our large dataset calculated $P_{99}(\dot{\varphi})$ but not $P_{99}(\dot{\varphi}u)$ or $P_{99}(\dot{\varphi}s)$, we approximate using $\dot{\varphi}_{99}U$, whose statistics are
nearly the same as $P_{99}(\dot{\varphi}u)$, also consistent with the relatively small skewness of $u$ exhibited during large $\dot{\varphi}$ events.





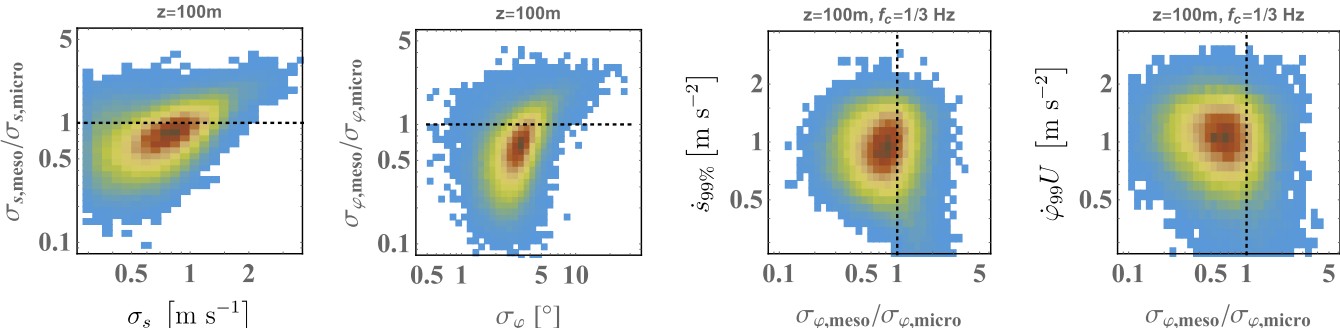

**Figure 10: Joint behavior (as jPDFs) with ratio of mesoscale (>2km) to microscale (<2km) directional variability. Left: ratio of respective mesoscale to microscale standard deviations versus $\sigma_s$ and $\sigma_\varphi$; right: $P_{99}$ of accelerations and corresponding ratio $\sigma_{\mathrm{meso}}/\sigma_{\mathrm{micro}}$. Dotted line indicates $\sigma_{s,\mathrm{meso}}/\sigma_{s,\mathrm{micro}} = 1$ and $\sigma_{\varphi,\mathrm{meso}}/\sigma_{\varphi,\mathrm{micro}} = 1$.**


In contrast to $\dot{s}_{99}$ or streamwise accelerations, $\dot{\varphi}_{99}$ does not exhibit a wind speed dependence; however, the directional accelerations and $\dot{\varphi}_{99}U \approx \dot{v}_{99}$ do have a dependence on $U$, analogous to that of $\dot{s}_{99}$ (and presumably $\dot{u}_{99}$). This is shown in the two left-hand plots of Figure 11 for $z = 160$ m and $f_c = 0.1$ Hz, with the same behavior also observed at 100 m height for all $f_c$ (not shown). In essence, the mode and commonly observed values of $\dot{\varphi}_{99}U$ (and thus $\dot{v}_{99}$) increase linearly with

mean wind speed $U$, while extreme values of $\dot{\varphi}_{99}U$ lack any dependence on wind speed. The right-hand plot of Figure 11 also shows that, analogous to the joint behavior of $\dot{s}_{99}$ with $\sigma_s$ and $\sigma_{s,\mathrm{micro}}$, the envelope of possible lateral accelerations represented by $\dot{\varphi}_{99}U$ increase proportionally with the microscale part of standard deviation of direction, $\sigma_{\varphi,\mathrm{micro}}$ (less precisely with $\sigma_\varphi$, not depicted) with extreme $\dot{\varphi}_{99}U$ having a limited dependence on $\sigma_{\varphi,\mathrm{micro}}$ (or $\sigma_\varphi$), though possibly increasing with $\sigma_{\varphi,\mathrm{micro}}$ but limited by sampling similar to how $P(\dot{s}_{99}, \sigma_{s,\mathrm{micro}})$ was. In contrast to $\mathrm{Mo}\{\dot{s}_{99}|\sigma_s\}$ and its

relationship with $\sigma_s$, the most common directional accelerations and conditional mode $\mathrm{Mo}\{\dot{\varphi}_{99}U|\sigma_\varphi\}$ appear to be independent of $\sigma_\varphi$.



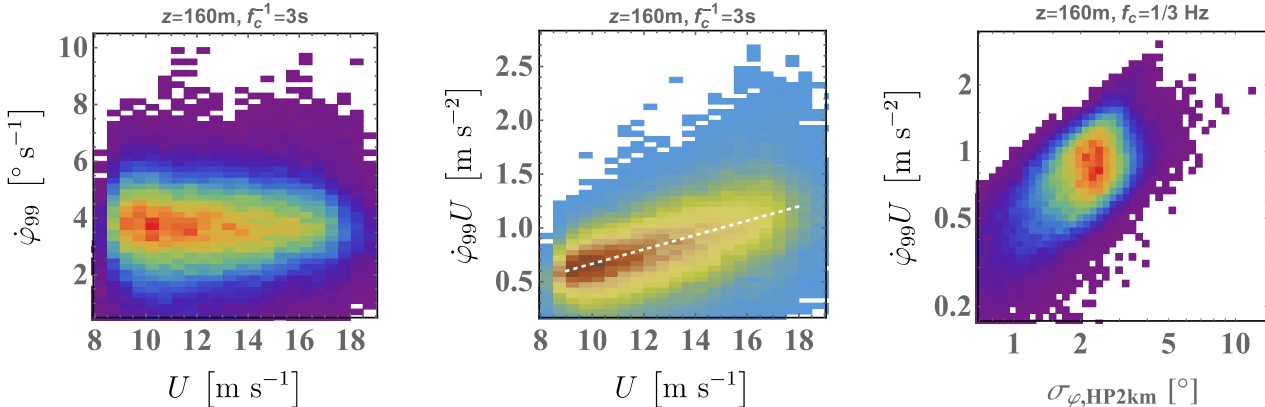

**Figure 11: Joint distributions of rate of change of wind direction and lateral acceleration with mean wind speed and microscale variability of direction (scales less than ~2 km). Dotted white line in $P(\dot{\varphi}_{99}U, U)$ shows linear relationship around the conditional**
**mode $\mathrm{Mo}\{\dot{\varphi}_{99}U|U\}$.**

The joint behavior of lateral and streamwise flow accelerations is shown in Figure 12, which gives $P(\dot{s}_{99}, \dot{\varphi}_{99}U)$ at both 100 m and 160 m heights, for $f_c =$1/10 Hz and 1/3 Hz. From the figure we see that for $f_c =$1/10 Hz the most commonly occurring conditions have $P_{99}$ of streamwise and lateral accelerations to be approximately equal, whereas for the higher filter
frequency of $f_c =$1/3 Hz we see for common conditions the lateral $P_{99}$ of accelerations exceed the streamwise. This is likely because inertial-range turbulence is not filtered out for $f_c =$1/3 Hz (with crosswind fluctuations being 4/3 larger than the lateral ones at the smallest scales). Using more severe low-pass filtering, with $f_c = $1/30 Hz (not shown) the opposite trend occurs and the most frequently occurring conditions have larger $\dot{s}_{99}$ than $\dot{\varphi}_{99}U$. Regarding the less common and extreme acceleration values, the plots in Figure 12 are made with linear axes to illustrate how the variability load-driving
accelerations increases; but we note that when plotted using log-log axes, the joint PDFs resemble those in Figure 6, Figure 7 and Figure 9; i.e., the joint variability around the mode (width of the jPDF envelope perpendicular to the 1:1 line) is relatively constant in log-space, scaling geometrically and consistent with log-normal distributions. The extreme accelerations do not exhibit a clear trend, but one can see that a fraction of extreme events involve both streamwise and lateral components. Further, comparing the extreme values between the plots of Figure 12 one can see that for different $f_c$
(again corresponding to different turbine/controller response times) then different events comprise the extremes. Some extremes with durations shorter than 10 s appear for $f_c = $1/3 Hz but are filtered out for $f_c = $1/10 Hz, particularly streamwise events without significant lateral accelerations. Recalling that the lateral 10-minute $P_{99}$ of acceleration $\dot{v}_{99}$ may be approximated by $\dot{\varphi}_{99}U$ and analogously the streamwise $\dot{u}_{99}$ approximated by $\dot{s}_{99}$, we point out that the "missing" pieces ($\dot{v}_{99} - \dot{\varphi}_{99}U$ and $\dot{u}_{99} - \dot{s}_{99}$, respectively) are not only small, but also behave similarly, so that a bias is not expected in the
joint variations shown in Figure 12. In the next section we will show how this looks for a sample joint extreme event, along with the strongest flow acceleration events measured over the 15 years of observations. Since three-dimensional sonic



anemometer recordings were available only at 80 m for most of the measurement period, our extensive set of calculations were made using $\dot{\varphi}_{99}$ and $\dot{s}_{99}$; future work includes re-calculation to obtain filtered $\dot{u}_{99}$ and $\dot{v}_{99}$ via combination of the anemometer and wind vane measurements, as these cannot be obtained through post-processing of the data used and results reported here.

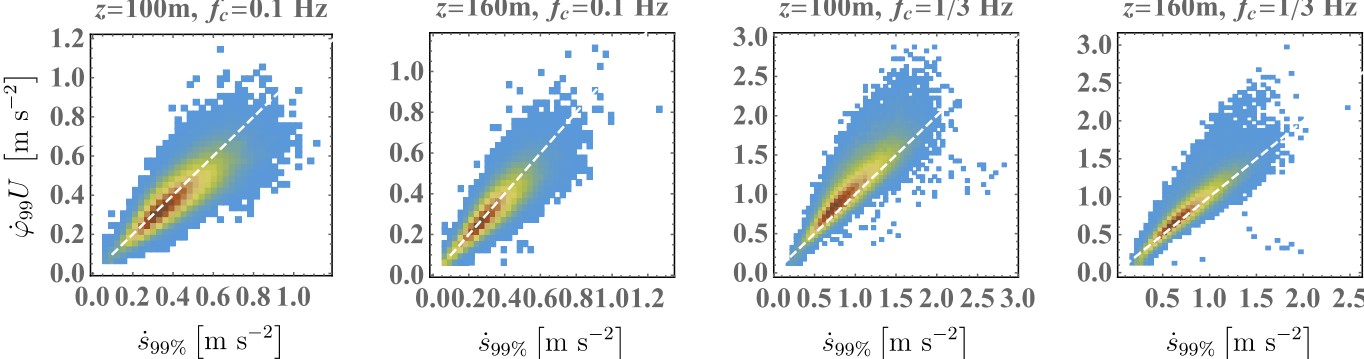

**Figure 12: joint PDF of 99$^{\text{th}}$ percentile of directional and horizontal flow accelerations, calculated as $\dot{\varphi}U$ and $\dot{s}$, respectively; plotted at both 100 m and 160 m heights, for characteristic response times ($f_c^{-1}$) of 10 s (left) and 3 s (right). Note linear axes are used here; log-log plots give jPDF shapes similar to those shown in Figs. 6,7, and 9. Dashed white line indicates 1:1 relation.**

### 3.3 Extreme flow accelerations

#### 3.3.1 'Anatomy' of an extreme event

To gain insight into what happens during an extreme acceleration event, we examine the acceleration components and wind speed together during a period containing such an event. Figure 13 shows the 10-minute segment for which the largest $\dot{s}_{99}$ with low-pass filter scale of 0.1 Hz was found for 10-minute mean speeds in the range 11–12 m s$^{-1}$ at 100 m height, i.e. $\max\{\dot{s}_{99}|(11 \leq U < 12 \text{ m s}^{-1})_{f_c=0.1 \text{ Hz}}\}$; this was also the second strongest $\dot{s}_{99}$ in this $U$-bin for $f_c = 1/3$ Hz, and the plots in the figure show results using $f_c = 1/3$ Hz. This extreme event was chosen not only due to its magnitude, but also because it happened during a limited time for which additional concurrent data was available: three-dimensional sonic anemometer data from 100 m and 160 m height on the same mast (denoted "LM," which hosts the cup anemometers and wind vanes whose data we have presented thus far), as well as at $z$=80 m from a second mast ("MM") located 400 m to the south. The upper-left plot of Figure 13 shows the "path" of $\{\dot{u}, \dot{v}\}$, i.e. the evolution of vector acceleration, measured by 3d sonic anemometer with sample rate $f_s = 20$ Hz. For comparison the upper-middle plot of the figure shows $\dot{s}$ with the directional acceleration (as $\dot{\varphi}U$) calculated using the same 3d anemometer. Several extreme flow accelerations are seen to occur, ranging from lateral to streamwise relative to the 10-minute mean wind direction, with $\dot{u}$ and $\dot{v}$ having similar maximum amplitudes. Comparing the $\{\dot{u}, \dot{v}\}$ and $\{\dot{s}, \dot{\varphi}U\}$ paths we see they are nearly identical, with slight distortions in the





lateral estimate $\dot{\varphi}U$ as expected and discussed following (4).[13] The largest excursions only indirectly affect the 99$^{th}$ percentile values of filtered accelerations, since $\dot{s}_{99}$ corresponds to the 120$^{th}$ largest value of $\dot{s}$ in a 10-minute period for $f_s = 20$ Hz (and 60$^{th}$ largest value for cup anemometers with $f_s = 10$ Hz); while the maxima of filtered $\dot{s}$ and $\dot{u}$ exceed 3 m s$^{-2}$ for the case shown in the figure, $\dot{s}_{99} = 1.6$ m s$^{-2}$ and $\dot{u}_{99} = 1.5$ m s$^{-2}$ for $f_c = 1/3$ Hz. The upper right-hand plot of Figure 13 displays the evolution of filtered acceleration with speed $s$, given as a joint-PDF to additionally indicate the distribution of speeds during the 10 minutes; the path of $\{s, \dot{s}\}$ evolves counterclockwise ($\dot{s} > 0$ leads to increasing speed, $\dot{s} < 0$ to decreasing speed). We also see that $\dot{s} > \dot{s}_{99}$ occurs across a range of wind speeds from ~8–15 m s$^{-1}$, with the largest $\dot{s}$ involving a jump from ~8 to 12 m s$^{-1}$. In this 10-minute period having $\sigma_s = 1.5$ m s$^{-1}$, the speed varies far beyond the 11-12 m s$^{-1}$ range defining the conditional mean; $s < 11$ m s$^{-1}$ for nearly half the period, and $s > 13$ m s$^{-1}$ for more than one minute, with $s$ repeatedly crossing typical rated speed (ca. 12 m s$^{-1}$ for multi-megawatt turbines). This is further illustrated in the middle plot, which displays the timeseries of speed and vertical velocity at 100 m and 160 m. It shows that similar speeds (and ranges of $\dot{s}$) occur at 100 m and 160 m heights, though sometimes $\dot{s}|_{z=160m}$ has the opposite sign as $\dot{s}|_{z=100m}$; this is related to significant vertical motions ($|w| > 1$ m s$^{-1}$) occurring at both heights, with a time lag (along with small direction changes). The middle plot also displays $s(t)$ from a cup anemometer at 100 m height on the same boom as the sonic anemometer, separated by about 5 m in the N-S direction; from it we see that the cup records nearly the same speed and does not exhibit the "overspeeding" that can occur due to large lateral velocity variance for some cup anemometers (Kristensen, 1998), despite the occurrence of large cross-wind accelerations, including some where $\dot{v} > \dot{u}$.

---

[13] We remind that this article focuses on streamwise extremes via $\dot{s}$: cup anemometers are industrially the most used type, and the processing of $\dot{s}$ from a cup is simpler than combining filtered signals from a cup and wind vane mounted with some distance between them. Again, the evaluation of $\dot{v}_{99}$ and its extremes is the subject of ongoing work.







**Figure 13:** "anatomy" of a 10-minute period of extreme $\dot{s}_{99}$ at height of 100 m with low-pass filter scale $f_c$ =1/3 Hz. Top left: path of horizontal acceleration vector from sonic anemometer; top center: path of $\{\dot{s}, \dot{\varphi}U\}$ from sonic; top right: path of $\dot{s}$ and speed $s$, plotted as joint PDF; center: timeseries of low-pass filtered speed and vertical velocity component (sonic at 100m is purple, sonic at 160m is orange, cup at 100m is dotted gray); bottom left: $\dot{s}$ at 100m height from cup and sonic anemometers; bottom center: $\dot{s}$ from two masts separated by ca. 400m, with $z$=80m and $z$=100m; bottom right: filtered acceleration components $\{\dot{u}, \dot{w}\}$ at 100m.

The lower-left plot of Figure 13 shows filtered $\dot{s}$ from the cup and sonic anemometers at 100 m, and their evolution together indicates that both anemometers are essentially measuring the same magnitudes of filtered acceleration, though with slightly larger $|\dot{s}|$ from the cup for the most extreme $\dot{s}$ that occurred around $t \simeq 200$–210 s. The correlation function between



them gives no persistent time lag, which is consistent with the anemometer separation being perpendicular to the wind direction of ~270°±15° recorded for this case; the flow structures passing the mast appear to have lateral dimensions greater

than 5 m and pass the sensors at $z = 100$ m simultaneously. The lower-middle plot of Figure 13 shows the mutual 'path' of filtered $\dot{s}$ from the sonic anemometers at 80 m height from the mast 400 m to the south ('MM') and at $z = 100$ m on the main mast ('LM') used thus far. The accelerations are effectively uncorrelated, though a persistent cross-correlation ($\rho > 0.6$) between the speeds is found for lags of ~50–100 s, suggesting that the flow structures have lateral extents of at least 400 m and yet larger streamwise extent (via Taylor's hypothesis), possibly propagating at an angle relative to the mean wind or

evolving at different rates in the crosswind direction. The existence of significant vertical accelerations is shown in the lower-right plot of Figure 13 from the sonic anemometer at 100 m, with the extreme magnitudes of $\dot{w}$ exceeding those of $\dot{u}$; these are essentially uncorrelated statistically, but one can see from the $\{\dot{u}, \dot{w}\}$ plot that during the extreme jumps in wind speed they are sometimes related (with varying lag), also noting such from the timeseries.

**3.3.2 Long-term extreme statistics: a collection of phenomena**

The 10 most extreme acceleration magnitude events were detected per each 1 m s⁻¹ increment of 10-minute mean wind speed between 8–17 m s⁻¹, at both 100 m and 160 m heights; i.e., 180 streamwise acceleration events (10 events × 9 speed "bins" × 2 heights) were found, with the corresponding 10-minute timeseries of speed and direction saved for use in constrained turbulence simulations to drive aeroelastic loads calculations (McWilliam *et al.*, 2023a).[14] The timeseries $s(t)$ for the most

extreme $\dot{s}_{99}$ in each $U$ 'bin' are shown in Figure 14, for a low-pass filter frequency of 0.1 Hz.

---

[14] Similarly, 180 directional extreme events were found; however, their analysis is left for subsequent work due to the recalculations needed, as mentioned in Sect. 3.2.



**Figure 14: pairs of 10-minute timeseries $s(t)$, for the most extreme acceleration events in each 1 m s$^{-1}$ wind speed bin for low-pass filter scale of 0.1 Hz. Left: 100m events (purple $s|_{100m}$, dashed-orange $s|_{160m}$); right: 160 m events (blue $s|_{160m}$, dashed-red $s|_{100m}$).**

The most extreme events detected at 100 m were different than those at 160 m, and the top $\dot{s}_{99}$ periods detected with $f_c$ =0.1 Hz are generally different than those found when $f_c$ =1/3 Hz (not shown). However, we note that for a given $U$ the most extreme event at one height and $f_c$ is often one of the most extreme events found for another $\{z, f_c\}$, such as the case shown previously in Figure 13. As can be inferred from the timeseries shown in Figure 14, a number of different qualitative properties and corresponding meteorological phenomena are associated with these extreme events. First, for some of the





100 m events (left-side plots in Figure 14) one can see that the corresponding 160 m speeds are either constant or look like a smoother 'version' of $s(t)$ at 100 m, particularly for smaller mean speed bins (below typical rated speeds). These correspond to shallow ABL depths that can occur during winter or nighttime in non-tropical climates (e.g., Liu & Liang, 2010; Kelly *et al.*, 2014), where the anemometer at 160 m height is near or within the stable inversion where turbulence is suppressed. The cases where $s_{160\,m}$ follows $s_{100\,m}$ tend to correspond to being just below the inversion, associated with

breaking gravity waves or wave-turbulence (Finnigan *et al.*, 1984; Einaudi & Finnigan, 1993) as in the top case for $f_c$ =0.1 Hz in the wind speed range $8 < U_{100\,m} \le 9$ m s$^{-1}$ or entrainment-zone turbulence (Otte & Wyngaard, 2001) as in the $14 < U \le 15$ m s$^{-1}$ cases for both heights[15] with $f_c$ =0.1 Hz (Figure 14). The cases with 'flat' $U_{160\,m}$ likely correspond to shallow ABL depths below 160 m having sufficiently strong capping inversions such that turbulence is suppressed. At higher speeds (above ~11 m s$^{-1}$) the extreme acceleration events at 100 m also tended to be accompanied by significant

accelerations at 160 m, which is consistent with the irregular spatial structure of the ABL-capping inversion (Sullivan *et al.*, 1998). Unfortunately, consistent ceilometer data from Høvsøre were not available to quantify the ABL depths during these periods with extreme $\dot{s}_{99}$.

For the top extreme events detected at 160 m (right-hand plots of Figure 14), the acceleration-associated jumps in wind speed were mostly accompanied by similar fluctuations at 100 m, consistent with ABL depths near 160 m. Further, for some

of these events one can also see steady winds before or after the jumps, commensurate with inversion depths fluctuating across this height. For several speeds there was also non-stationarity at both 100 m and 160 m occurring during daytime hours, consistent with frontal passage; this included two wind speed ramp events.

Further identification of these events and confirmation of their driving mechanisms may be accomplished through analysis of mesoscale simulations for the site, which is left for future work. Timeseries of the top 10 events in each mean-speed bin

for $f_c = 0.1$ Hz were provided to and used by McWilliam *et al.* (2023b) for aeroelastic simulations, along with Mann-model turbulence parameters corresponding to each respective period found from the cup-vane combinations via a new method (see Appendix A). The timeseries in that dataset were provided as 1 Hz (down-sampled from 10 Hz) records of speed, direction, streamwise velocity component, and lateral velocity component, at both 100 m and 160 m heights; the reader may obtain these from the reference dataset of McWilliam *et al.* (2023a).


### 3.4 Load-driving accelerations, from fatigue to ultimate

Towards practical use and enabling comparison with wind turbine standards, the behavior of dominant filtered accelerations have been examined via the top 1% per every 10-minute period, i.e., using $\dot{s}_{99}$ and $\dot{\varphi}_{99}$; this includes statistics conditional on the mean wind speeds $U$ and the corresponding standard deviation ($\sigma_s$ or $\sigma_\varphi$), as described in the previous sections. The most

---

[15] Note offshore convective cells could give a similar signature in windspeed timeseries (e.g., Agee, 1987; Vincent *et al.*, 2012). However, these were ruled out via concurrent $w(t)$ available from 3d sonic anemometer for these cases (not shown), which gave no evidence of vertical motions associated with cellular convection.





common values of horizontal (streamwise) and directional (lateral) 99$^{th}$-percentile accelerations, calculated here via $\dot{s}_{99}$ and $\dot{\varphi}_{99}s$, are presumably what drive some fatigue loads on wind turbines, particularly thrust-based loads such as flap-wise root bending moment and tower base fore-aft moment (Frandsen, 2007; Kelly *et al.*, 2021). Although the most commonly occurring $\dot{s}_{99}$ have been shown above to be analytically describable through the conditional modes $\text{Mo}\{\dot{s}_{99}|U\}$ and $\text{Mo}\{\dot{s}_{99}|\sigma_s\}$, current industrial practices and the IEC 61400-1 standard already prescribe fatigue-testing design load cases

(DLCs) in terms of $U$ and $\sigma_s$. Due to the latter, since we find that $\text{Mo}\{\dot{s}_{99}|U\}$ and $\text{Mo}\{\dot{s}_{99}|\sigma_s\}$ monotonically follow the behavior of $U$ and $\sigma_s$, while $\text{Mo}\{U\dot{\varphi}_{99}\}$ is independent of $\sigma_\varphi$ which is ignored by the standard, then employing acceleration statistics for fatigue loads might have limited usefulness. This is underlined also by the most common $U\dot{\varphi}_{99}$ linearly following $\dot{s}_{99}$ while the IEC 61400-1 (§6.3.1) also prescribes the lateral turbulence strength $\sigma_v$ to be proportional to the streamwise ($\sigma_u$), though the ratio of $\sigma_v$ to $\sigma_u$ prescribed for fatigue DLC's is different than the ratio of most common

$U\dot{\varphi}_{99}/\dot{s}_{99}$ found here. The latter aspect, and specifically $\dot{v}_{99}/\dot{u}_{99}$, is the subject of future work (as more extensive calculations are needed for such). The magnitudes of extreme lateral acceleration estimates found here were also used by Hannesdóttir *et al.* (2023) in aeroelastic simulations for coherent gusts having extreme directional changes, who determined that they induce loads much weaker than those arising from the 61400-1 standard's prescriptions. Because of this, we do not further pursue lateral extremes here, leaving for future work more accurate calculation of them via statistics of explicitly

calculated $\dot{v}_{99}$ (via eqn. 3, not approximations such as $s\dot{\varphi}_{99}$ or $U\dot{\varphi}_{99}$).

On the other hand, the extreme $\dot{s}_{99}$ were shown above to have behavior differing from IEC prescriptions, notably lacking a discernable dependence on wind speed or a strong correlation with $\sigma_s$. We remind that Kelly *et al.* (2021) found wind speed ramps crossing rated speed with $\Delta s/\Delta t$ near 0.5 m s$^{-2}$ can exceed DLC1.3 of the 61400-1, and this acceleration magnitude is smaller than the top tenth of $\dot{s}_{99}|_{f_c=0.1\,\text{Hz}}$ values found here for speeds above 11–12 m s$^{-1}$. Further, McWilliam *et al.* (2023b)

performed a Monte Carlo set of aeroelastic simulations driven by constrained turbulence according to the $\dot{s}_{99}$ reported here, and found some modelled loads exceeded the 64100-1 prescriptions. Thus the $\dot{s}_{99}$ extremes are worth further consideration.

For consideration of extreme accelerations and their potential effects on loads and control, we return to the long-term statistics of $\dot{s}_{99}$, reminding that the largest $\dot{s}_{99}$ were found in Section 3.1 to follow a log-normal distribution (2) with parameters shown in Table 2. The cumulative distribution function (CDF) is obtained by integrating (2); then inverting this

CDF, the value $\dot{s}_{99}$ corresponding to a given value of the CDF, also called the quantile $q$, can be expressed as

$$\dot{s}_{99}|_e(q) = \exp\left[\mu_e - \sqrt{2}\,\sigma_e\,\text{erf}^{-1}(1-2q)\right] \qquad (6)$$

where $\text{erf}^{-1}(1-2q)$ is the inverse of the error function[16] (Abramowitz & Stegun, 1972) evaluated at $1-2q$. Accounting for the fraction of observations covered by the range of speeds considered and the total length of the dataset, noting also that $q = 1 - T_0/T_{\text{ret}}$ for a base period $T_0$ and return period $T_{\text{ret}}$ we can use (6) to get the $\dot{s}_{99}$ expected for a given $T_{\text{ret}}$. Doing so,

from the parameters listed in Table 2 we can then estimate the expected 10-minute $\dot{s}_{99}$ for a given filter scale (turbine

---

[16] Note the inverse error function expression can be written more compactly in terms of the inverse of the complementary error function, i.e., $\text{erf}^{-1}(1-2q) = \text{erfc}^{-1}(2q)$; this is sometimes reported in the literature instead.




response time), over longer periods; this is displayed in Figure 15, which gives the expected $\dot{s}_{99}$ for the three low-pass filter frequencies (characteristic turbine response times) and two heights considered. The primary lines represent a base period of 39 minutes, equal to 10 minutes scaled by the rate of occurrence of directions considered within the range of speeds analyzed divided by the fraction of data satisfying the selection criteria outlined in section 2; i.e., it is the ratio of total timespan to the

number of samples used, scaled by the fraction of winds within $8 < U < 18$ m s$^{-1}$ that occur from offshore directions.

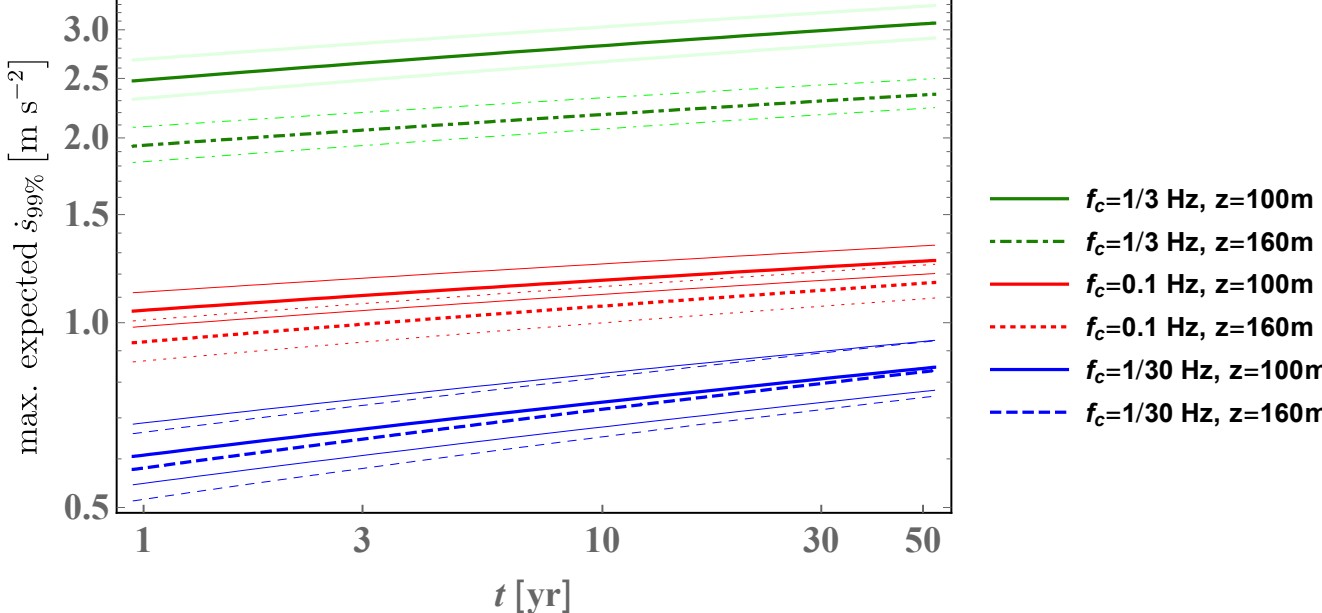

**Figure 15: maximum filtered acceleration expected per return period, for the three different filter scales considered and two heights analyzed. For each case, thick lines correspond to $T_{\text{base}} \simeq 39$ minutes (total timespan divided by number of samples); shaded areas and thin lines around each case show range for base-periods varying from 10–119 minutes (see text for explanation).**

To demonstrate the largest possible variation due to definition of the base period $T_0$, Figure 15 also shows bands around each line, which may overlap and are thus bounded by thin lines corresponding to the style of each case (e.g., dotted for $f_c = 0.1$ Hz at $z = 160$ m). The bands show the range of expected extreme $\dot{s}_{99}$ resulting from base periods ranging from 10 minutes to 119 minutes, where the latter corresponds to neglect of the directional rate of occurrence. For the shortest response timescale (highest filter scale, $f_c = 1/3$ Hz) we see a variation of about ±6% for 50-year $\dot{s}_{99}$, but note that this is

the upper limit of uncertainty expected due to base-period representativity. One sees quite dramatically that higher $f_c$ (shorter turbine reaction timescales) give significantly larger 99$^{\text{th}}$-percentile flow accelerations, more than a factor of two when comparing $f_s$ of 1/3 Hz and 1/10 Hz. Further, a larger difference is seen between $z=100$ m and $z=160$ m for $f_s = 1/3$ Hz compared to slower response times (higher $f_c$), with stronger $\dot{s}_{99}$ at $z=100$ m, due to significant accelerations having characteristic sizes of roughly 30–50 m; more analysis needs to be done at higher $z$ to determine if this trend reverses.

We remind that the expected extreme accelerations in Figure 15 correspond to the range of wind speeds (8–18 m s$^{-1}$) at the two heights ($z = 100$ m, 160 m) considered; there was no apparent dependence on wind speed as shown in Figure 5, though



the largest $\dot{s}_{99}$ occurred for $10 < U \lesssim 15$ m s$^{-1}$, which coincides with the most common 10-minute mean wind speeds observed around the long term mean (at 100 m the most common speed was 9.6 m s$^{-1}$, the mean was 10.6 m s$^{-1}$, the CDF at 15 m s$^{-1}$ was 0.83). We do not calculate contours of expected 50-year $P(\dot{s}_{99}, U)$, due to not yet having analyzed cases below

8 m s$^{-1}$ or above 18 m s$^{-1}$, and because the directional limitations of this coastal site limited the overall number of offshore winds sampled; this causes difficulty in fitting conditional extreme distributions, and larger uncertainty relative to finding the marginal distribution. A larger offshore dataset would be needed to make such two-dimensional 50-year contours.

## 4 Discussion and conclusions

From all low-pass filtered 10-minute $\dot{s}_{99}$ found for offshore flow over a 17-year period at the coastal Høvsøre site, the largest observed flow accelerations correspond to events having durations longer than the reciprocal of filter scales chosen ($f_c^{-1}$). These are long enough to significantly affect wind turbine loads for turbines with characteristic controller/response times of $f_c^{-1}$ (or shorter), if the transverse spatial scales of the flow structures are sufficiently large. Invoking Taylor's hypothesis to get a crude estimate, the streamwise length scales would be on the order of the product of mean speed and duration, giving

gust widths of ~25–35 m for $f_c^{-1} = 3$ s and beyond 250 m for $f_c^{-1} = 30$ s; for roughly isotropic disturbances, assuming the transverse extent is similar, this is easily large enough to affect conventional turbine blades and associated thrust-based loads. Further, for some extreme acceleration-inducing flow mechanisms, such as cold-front passage or breaking gravity waves associated with the capping inversion, one expects the transverse extent to be much longer than the streamwise one. Larger $\dot{s}_{99}$ tend to correspond to shorter event durations, with larger amplitude $\dot{s}_{99}$ associated with higher $f_c$ (faster turbine

response); this implies length scales roughly as small as the minimum gust widths noted above, thus for typical offshore turbine blade lengths (> ~50 m), for effective turbine response times of $f_c^{-1} = 3$ s, the shortest extreme gusts' effect on loads could either be mitigated somewhat (e.g., as in load-shedding approaching rated speed via fast controller) or possibly induce larger blade loads (e.g., flap-wise bending moments) due to a single blade being impacted.

It is remarkable that the growth of wind turbines in recent years — both hub heights and blade lengths — has not only

caused offshore turbine loads to become increasingly impacted by upper-ABL phenomena (more so than surface-induced turbulence), but additionally that the *character* of extreme events has changed, due in part to the different physical scales of extreme transients above the marine surface layer. Also notable is the variety of diverse signatures exhibited by the extreme acceleration events at 100 m and 160 m heights. The identified events indicate both turbulent and non-turbulent flow regimes, including some associated with the stable capping inversion above shallow ABLs; the latter include phenomena

such as breaking gravity waves, wave-turbulence, entrainment outbreaks, and 'top-down' intrusions, while we also noted extremes associated with frontal passages and other phenomena that have limited association with the ABL-top (e.g., borders between strong coherent structures).

Statistically, stronger filtered accelerations were found at 100 m height compared to 160 m, across all wind speeds and including the extremes, for the offshore flow considered at Høvsøre. One might expect interaction with the sea surface to be



responsible for this, but the most extreme events are not generally turbulent, especially for $f_c \leq 1/10$ Hz; for $f_c = 1/3$ Hz some (more) turbulence is seen, implying at higher $f_c$ (faster response times) the surface may have more impact. The effect of the strip of land between mast and the ocean is also irrelevant at these heights, as choosing to analyze speeds above 8 m s$^{-1}$ also removes significantly unstable conditions (which could otherwise cause ground effect via mixing). Further investigation at more offshore sites and heights can help clarify this aspect. Though speeds from ~6–8 m s$^{-1}$ are moderately

common, 10-minute mean wind speeds below 8 m s$^{-1}$ were ignored because transients at lower speeds (further from rated speed) generally have less impact on loads (Dimitrov *et al.*, 2018; Kelly *et al.*, 2021), and to avoid the coastal effects in unstable conditions; this is further justified by our finding that extreme accelerations for $U \lesssim 11$ m s$^{-1}$ are appreciably smaller than those with $U \gtrsim 11$ m s$^{-1}$ (Figure 5), while mean speeds between $8 \lesssim U \lesssim 11$ m s$^{-1}$ occur more frequently than those above 11 m s$^{-1}$. If we included lower speeds and a dependence of extreme $\dot{s}_{99}$ decreasing at smaller $U$, then use of the

marginal extreme distribution and associated statistical extrapolation (Figure 15) would give lower predictions of extreme acceleration at mean wind speeds below ~11 m s$^{-1}$. No $U$-dependence was found for extremes approaching the high end of speeds analyzed (18 m s$^{-1}$, again from Figure 5), where less impact is expected from the transient accelerations due to being above rated speed; although shut-down cases could be considered for comparison with the IEC 61400-1, much more data would be needed to investigate these due to the relative rarity of speeds crossing above cut-out (around 25 m s$^{-1}$).

We have assumed that the (westerly) conditions analyzed are representative for all directions offshore, but one could conceive that the significant south-easterly winds which sometimes occur in the spring at Høvsøre (Peña *et al.*, 2016) could be different enough to have a small impact on the statistics; however, such wind directions are even less common offshore in the North Sea and North Atlantic wind climates characterized by the measurements (e.g., Hahmann *et al.*, 2022). We can further get a sense of the limited potential impact on 50-year extreme accelerations, considering the 'error' lines in Figure

15, which represent neglect of the fraction of speeds not considered, and fraction of winds coming from offshore, respectively.

     This work has produced statistics for dominant flow accelerations detected using three different low-pass filter frequencies $f_c$ (as proxies for characteristic turbine response times), but yet more utility could be obtained by characterizing the systematic effect of low-pass filtering on extreme acceleration statistics; i.e., to find an explicit dependence of the

extreme $\dot{s}_{99}$ distribution on $f_c$. Attempts were made to this end, but not included because no simple relation was not found to fit the data. Scaling $\dot{s}_{99}$ by $f_c^{-0.6}$ collapsed the most extreme filtered acceleration amplitudes (with SF$<10^{-4}$ in Figure 4) to a single curve for both $f_c = \{1/3, 1/30\}$ Hz at $z = 100$ m and for $f_c = \{1/3, 1/10\}$ Hz at $z = 160$ m (not shown), but no $f_c$-scaling can collapse the extreme distributions or survival functions for all three filter frequencies. This is not surprising: again, most of the extreme events are not 'simply' due to inertial-range turbulence (which permits a simple scaling) or any

single phenomenon, though we note more turbulence is observed at 100 m than 160 m during extreme events. The relative rates of occurrence and relative variation in the strength of the phenomena causing extreme load-driving accelerations is seen to depend not only on $f_c$, but also on distance to both the ground and to the capping inversion, as well as the capping inversion strength (Pedersen *et al.*, 2014; Kelly *et al.*, 2019).





For use in Monte-Carlo aeroelastic simulations, a method was developed (Appendix B) to employ the extreme distributions of offshore filtered acceleration derived above. Practical stochastic expressions are given to relate the magnitude and duration (gust rise time) of filtered flow acceleration including rise time distributions, applicable within the IEC 61400-9 or as a probabilistic supplement (replacement) for the EOG prescription found in the IEC 61400-1; we note these practical expressions followed from earlier wind speed ramp acceleration studies, and the exact constants and forms may be improved with further investigation and analysis. Additionally considering the IEC 61400-1, its EOG prescription has an implicit rise time of almost 3 s, and for contemporary wind turbines in the highest turbulence subclass (A+), around rated speed it implies characteristic event acceleration magnitudes that are similar to the 50-year values obtained from measurements at 100–160 m heights with $f_c =1/3$ Hz. For higher $V_{\mathrm{hub}}$ the IEC EOG prescription gives larger accelerations than 50-year $\dot{s}_{99}$ found here from 100–160 m observations with $f_c =1/3$ Hz (even for different turbulence subclasses), and $\dot{s}_{\mathrm{EOG}}$ are generally larger than 50-year $\dot{s}_{99}$ from measurements with $f_c =0.1$ Hz or 1/30 Hz; the lowest turbulence subclass ($I_u = 12\%$) gives weaker accelerations than 50-year $\dot{s}_{99}|_{f_c=1/3\,\mathrm{Hz}}$ for speeds below rated (Appendix B). We note that the IEC 61400-1 standard — due to its original basis onshore and with $z_{\mathrm{hub}}$ closer to the surface — prescribes its EOG in terms of turbulence intensity, which is not realistic for offshore turbines with typical hub heights beyond 100 m; as we have seen, load-driving flow accelerations do not follow 10-minute standard deviations of wind speed or velocity.

We remind that the results and conclusions herein are for offshore wind; over land, the dominance and effect of turbulence extends further from the surface, typically beyond hub height (e.g., Alcayaga, 2017).

## 4.1 Outlook and continuing work

Continuing work includes further translation to probabilistic gust definitions for the IEC standards, whereby joint distributions of extreme flow accelerations and associated rise times (or magnitude of speed increase) facilitate an update of the extreme operating gust ("EOG") in the 61400-1 as well as prescriptions supporting Monte Carlo aeroelastic simulations for the 61400-9. From the acceleration statistics found here, I derived an offshore probabilistic gust prescription towards the IEC 61400-9 standard, as given in Appendix B. More explicit systematic quantification of the durations associated with extreme flow accelerations, with rise time statistics conditioned on wind speed and amplitude (analogous to that for ramps by Kelly, *et al*., 2021), is still ongoing; this should also be done for more offshore sites and heights. One aspect involves the relationship between extreme flow acceleration amplitudes and gust duration. Following ramp studies and preliminary analysis here we have taken extreme events to have $t_d \propto 1/\dot{s}_{99}$ in a statistical sense with $\Delta s \propto \dot{s}_{99} t_d$; though such events are due to different phenomena beyond turbulence, a physical hypothesis is that these extreme accelerations are (mostly) attributable to passage of a 'border' between coherent flow structures, with the fluid equations of motion and conservation of mass limiting $\Delta s$ and causing the inverse relation between extreme $\dot{s}_{99}$ and duration. More investigation is needed to explicitly determine the joint behavior of extreme $\dot{s}_{99}$ and the associated $\{t_d, \Delta s\}$, along with the extent to which the border-



zone width and advection speed determine the largest acceleration magnitudes. A related aim is to more directly measure the characteristic length scales and orientations of extreme flow acceleration events, through both mast-based anemometers and lidar, including more multi-point measurement statistics to characterize the associated flow structure(s). Doing so permits better modelling of transient forces on turbine blades and rotors, through constrained aeroelastic simulations incorporating the multidimensional length-scale information.

Starting with single-point statistics, further work could help quantify the behavior and joint distributions of $\{u, s\}$ for the most common conditions at heights of interest offshore (above the surface layer). Although we found here that extreme acceleration events have $\dot{s}_{99} \simeq \dot{u}_{99}$ while for the most commonly-occurring conditions $\dot{s}_{99} \propto \sigma_s$, for the latter case we cannot definitively yet state the degree to which $\dot{u}_{99} \propto \sigma_u$ though we do expect such. Classic turbulence theory gives ideal relations between $\{\sigma_u, \sigma_v, \sigma_w\}$ but and for fatigue loads the IEC 61400-1 uses such in its prescriptions; but, the use of measured $\sigma_s$ in place of $\sigma_u$ has not been directly addressed, though it might be implicitly accounted for within the empirical constants used in the standard. Further, while theoretical forms are also available relating $\{\sigma_{ds/dt}, \sigma_s, \sigma_{du/dt}, \sigma_u\}$, they do not necessarily apply for the non-ideal flow structures which behind the relationships between $\{\dot{s}_{99}, \sigma_s, \dot{u}_{99}, \sigma_u\}$ and thus fatigue loads. However, this has likely been approximately accounted for in effect by the IEC's empirical description using $P_{90}(\sigma_u)$; we do note that the latter does not deal with the tails of the PDF from each 10-minute period, in contrast with long-term statistics of $\dot{s}_{99}$ or $\dot{u}_{99}$. But the behavior of commonly-occurring $\dot{s}_{99}$ may not be markedly different than that of $\sigma_u$ in terms of its effect on for fatigue loads, which is what Figure 6 appears to imply; though this remains to be directly shown from observations, we expect the flow acceleration paradigm to be more important for extremes.

On the meteorological side, remaining work includes matching flow regimes and conditions to the observed extreme events via analysis of mesoscale model output (e.g. WRF) and extended observational data; this also includes exploration of ABL depths, whose observed values can differ from mesoscale model predictions. Such work allows for investigation of the flow mechanisms and meteorological phenomena that cause extreme flow accelerations offshore. It is also worth noting that microscale models such as LES are *not* expected to replicate extreme events like those observed — due to the need to know and simulate details such as the varying strength and structure of capping inversions, as well as variable mesoscale forcings. Coupled models such as WRF-LES might be able to reproduce *some* of the phenomena; however, the relative rates of occurrence of the various phenomena producing extreme flow accelerations, and the fidelity of simulation of such transients, is unknown, and follows *after* the non-trivial analysis of matching flow regimes to observed conditions.

Furthermore, the response of different turbine controllers to extreme accelerations should be examined. For example, at some speeds certain wind turbines may not be as affected by large accelerations as much as other turbines (depending on the duration and physical extent of the flow disturbance), in a way that is more complicated than captured by accelerations identified via simple low-pass filtering via $f_c$. It is possible that more representative filters (e.g., band-pass) can be made to find acceleration statistics based on common control strategies near rated power, and that other parameters (e.g., pitch angle) need to be considered. The analysis presented herein, as well as the ongoing work just mentioned, also needs to be expanded



to include speeds across the *full* range of wind turbine operation (beyond 8–18 m s⁻¹), to heights above 160 m, and at
multiple offshore (and potentially onshore) sites. Lastly, a mean speed dependence has not yet been found or incorporated
into the extreme flow acceleration statistics, but with more measurements this might be elucidated.

**Appendix A: method for estimation of Mann-model parameters from cup & vane measurements**

Only wind speed and direction data at 100 m and 160 m were available from cup anemometers and vanes for most of the
observational period, without three-dimensional velocity component data. Due to this, to facilitate use of extreme
acceleration timeseries (whether observed as in Sect. 3 or synthesized as in Appendix B), it was necessary to create a method
to practically derive Mann-model turbulence parameters from such measurements for use in constrained gust simulations —
as made by McWilliam *et al.* (2023b) based on the timeseries described in Sect. 3. Such a method for finding turbulence
parameters without vertical velocity components is also useful because 2D cup/vane instrumentation, or floating lidar[17], is
standard for industrial wind energy pre-construction measurement campaigns.
Although Kelly (2018) found the Mann-model turbulence length scale to be expressible in terms of shear through the bulk
relation $L_M \approx \sigma_s/(\Delta U/\Delta z)$, this stems from turbulence dominated by shear production, which is not the case for the flow
behind many extreme acceleration events; furthermore, we wish to address vertical inhomogeneity (variation of shear and
turbulence) and allow use of timeseries from different heights for constrained turbulence simulations that embed acceleration
events. We are thus challenged to diagnose the length scale using two-dimensional (horizontal) information at individual
heights, in a way that matches measured spectra from three-dimensional sonic anemometers.

Based on the limited 3D sonic anemometer measurements available at 100 m and 160 m, and independent sonic
measurements from the shorter mast at 80 m height, one-dimensional frequency spectra (thus streamwise wavenumber
spectra via Taylor's hypothesis) of $uu, uw, vv$ and $uw$ were calculated for the 10 most extreme acceleration events per wind
speed bin identified at each location. These allowed $\chi^2$ fits to the single-point velocity spectral tensor $\Phi_{ij}$ (Mann, 1994;
IEC, 2019), which give the three Mann-model turbulence parameters $(L_M, \alpha\epsilon^{2/3}, \Gamma)$ that allow synthesis of turbulence
timeseries following the 61400-1 standard (IEC, 2019). The flow during extreme events exhibited significant non-
stationarity due to mesoscale (low-wavenumber) variance which is not well-represented by Mann-model spectra[18], so the
turbulence with each event was also found from fits to spatially high-pass filtered spectra using Taylor's hypothesis and a
second-order Butterworth filter with $f_{hp} = U/(2\,km)$. From the raw and high-pass filtered three-dimensional velocity

---

[17] Floating lidar has become common in offshore pre-construction observation campaigns, but commercial profiling lidar and
data-loggers are typically configured in such campaigns to save only 10-minute averages of horizontal wind components;
their scanning patterns are not configured to capture transient acceleration events. Forward-looking nacelle-mounted lidars
also show promise, but they are not yet in widespread use, and it will take some years before they are able to measure
sufficient statistics (presuming relevant high-frequency long-term statistics are even saved for operating farms).
[18] Syed & Mann (2024) recently made a modification to the Mann model for handling low-wavenumber (large-scale)
fluctuations which may appear as non-stationarity, but such work happened after my analysis was completed.





timeseries it was found that the turbulence length scale could be estimated most simply by using the integral time scale of lateral velocity fluctuations: the raw signal (i.e., including non-stationarity) gave $L_M \approx 0.5 L_v \simeq 0.5 T_v U$, while the high-pass filtered turbulence had $L_{M,hp} \approx 0.8 T_{v,hp} U$, where the integral time scale is calculated through temporal autocorrelation and the subscript 'hp' denotes spatially high-pass filtered turbulence. The $L_{M,hp}$ estimates are much better than unfiltered $L_M$, with Figure 16 displaying joint PDFs of estimated versus observed (spectrally fit) $L_{M,hp}$ over the collection of extreme

acceleration events. One can see some scatter in the estimates, but we note that the acceleration (gust) description is the focus of constrained simulations, and the background turbulence length scale is secondary. We further remind that explicit calculation of the length- and timescales associated with extreme accelerations is ongoing/future work.

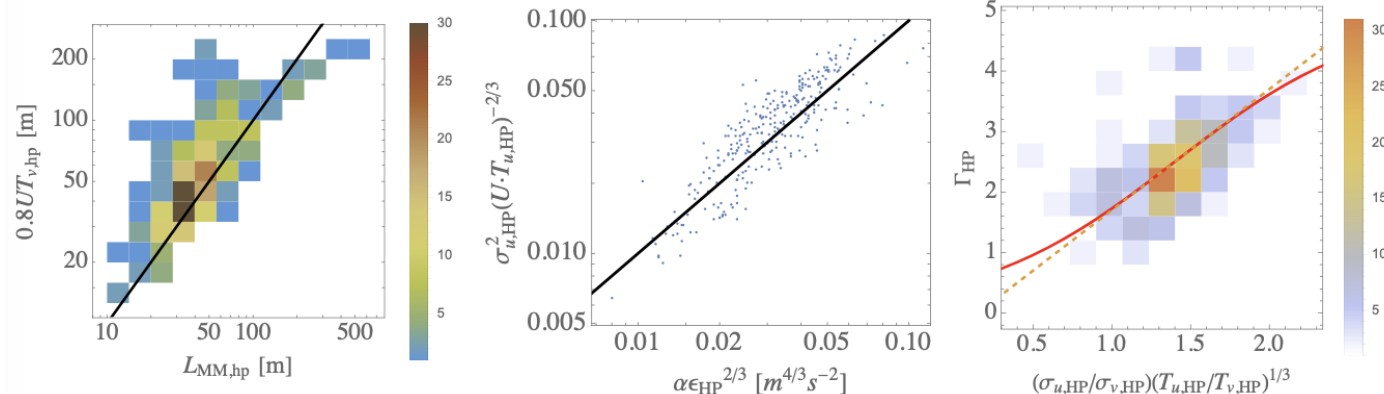

**Figure 16: estimates versus spectrally fit values of Mann-model parameters for spatially high-pass filtered ($f_{HP} = U/2$ km)**
**turbulence behind the extreme acceleration events. Left: length scale, with 1:1 indicated by black line. Center: amplitude parameter $\alpha \epsilon^{2/3}$, again with black 1:1 line. Right: anisotropy parameter Γ, with expression (A.2) as solid line and its linear equivalent as dashed. Colors in jPDFs indicate event count, to show how estimate is covering most commonly observed values.**

Analogous to obtaining $L_M$, it was found that the Mann-model amplitude parameter can also be estimated using the integral

timescale. Knowing $\alpha \epsilon^{2/3}$ scales as $\sigma_u^2 L^{2/3}$ (Mann, 1994; Kelly, 2018), from the fits to the spatially high-pass filtered spectral tensor components (again with $f_{hp} = U/2$ km), we find that

$$\alpha \epsilon^{2/3} \simeq \sigma_{u,hp}^2 (T_{u,hp} U)^{-2/3} \tag{A.1}$$

matches the three-dimensional spectrally-fit values well, and note that $\alpha \epsilon^{2/3} \approx 0.7 \sigma_{u,hp}^2 (T_{v,hp} U)^{-2/3}$ also gives a crude estimate. The estimate from (A.1) versus spectrally-fit $\alpha \epsilon^{2/3}$ are shown in the center plot of Figure 16 for the extreme

acceleration events.

Lastly, we find a form to estimate for the anisotropy (eddy-lifetime) parameter Γ, which we expect to be proportional to

$(T_u/T_v)^{1/3}$ and $\sigma_u/\sigma_v$ from Kelly (2018); we find that the most common values follow $\Gamma_{hp} \approx \left[ 2 \left( \frac{\sigma_{u,hp}}{\sigma_{v,hp}} \right) \left( \frac{T_{u,hp}}{T_{v,hp}} \right)^{1/3} - 0.3 \right]$,

whereas keeping the mesoscale fluctuations (not high-pass spatial filtering or 'de-trending') degrades the fits, and leaves



little discernible pattern for estimation of $\Gamma$ in terms of unfiltered quantities that we measure. Because $\Gamma$ cannot be negative
and is limited to $\Gamma \le 5$ (Mann, 1994), a practical form to estimate it is

$$\Gamma_{\mathrm{hp}} \approx 2.5 \left\{ 1 + \tanh\left[ 0.8 \left( \frac{\sigma_{u,\mathrm{hp}}}{\sigma_{v,\mathrm{hp}}} \left( \frac{T_{u,\mathrm{hp}}}{T_{v,\mathrm{hp}}} \right)^{1/3} - 1.4 \right) \right] \right\}; \tag{A.2}$$

this is shown in the right-hand plot of Figure 16, superposed on the joint PDF of $\left(\sigma_{u,\mathrm{hp}}/\sigma_{v,\mathrm{hp}}\right)\left(T_{u,\mathrm{hp}}/T_{v,\mathrm{hp}}\right)^{1/3}$ and $\Gamma$ from
Mann-model fits to spectra of $\{uu, uw, vv, uw\}$.

**Appendix B: use of extreme acceleration distribution for gust synthesis and simulation**

For investigating effects on turbine loads and probabilistic design (e.g., via Monte Carlo simulation), it is useful to be able to
'convert' the extreme acceleration statistics into gusts for simulation, as a probabilistic or alternative to (or augmentation of)
the IEC 61400-1 standard's extreme operating gust or 'EOG'. On average the expected duration or "rise time" for a given
streamwise extreme flow acceleration, denoted by $T_d$, tends to decrease with acceleration amplitude. This has been
documented in wind ramp studies (Hannesdóttir *et al.*, 2019ab; Kelly *et al.*, 2021), with the latter showing rise times to be
inversely proportional to acceleration; plotting the joint distribution $P(T_d, \Delta s/\Delta t)$ from the data of Kelly *et al.* (2021), as in
Figure 17, we more clearly see $T_d \approx (6\ \mathrm{m\ s^{-1}})/(\Delta s/\Delta t)$ for ramps.

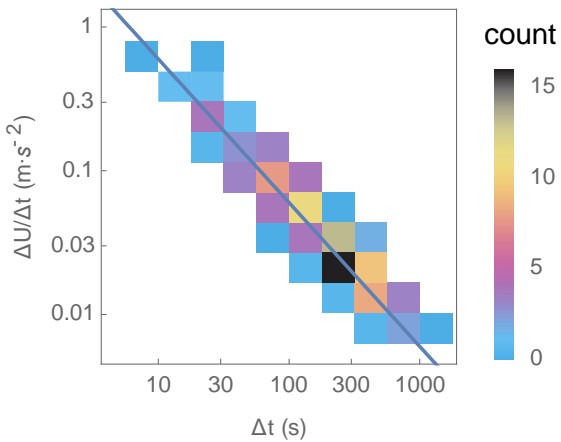

**Figure 17: joint PDF of bulk accelerations and rise times from the ramp data of Kelly *et al.* (2021). Solid line is $(6\ \mathrm{m\ s^{-1}})/\Delta t$.**

One could adapt this for extreme accelerations $\dot{s}_e$ having durations larger than a given filter timescale $f_c^{-1}$; however, we
observe that less extreme accelerations do not necessarily have such long durations[19], and that such a simple expression

---

[19] The characteristic time and length scales of ramps, and causes, are different; ramps tend to be caused by passage of (cold)
fronts, the strongest accelerations found here have shorter timescales and few are attributable to frontal passage.



would always give the same wind speed increase of $T_d \dot{s}_e = 6$ m s$^{-1}$. From preliminary analysis of the strongest events, for practicality we extend $T_d \approx (6$ m s$^{-1})/\dot{s}_e = \Delta s_{\text{ref}}/\dot{s}_e$ to suggest the form

$$T_d(\dot{s}_e) \simeq \frac{\Delta s_{\text{ref}}/a_{\text{ref}}}{[1+(\dot{s}_e/a_{\text{ref}})^\xi]^{1/\xi}} \; ; \tag{B.1}$$

here $a_{\text{ref}} \simeq 0.4$ m s$^{-2}$ is the value below which the durations are shorter than $\Delta s_{\text{ref}}/\dot{s}_e$, with $T_d \to \Delta s_{\text{ref}}/\dot{s}_e$ for extreme accelerations $\dot{s}_e > a_{\text{ref}}$. The coefficient $\xi = 3$ was found empirically along with $a_{\text{ref}}$, and we note the parameters in (B.1) could be refined through ongoing work, wherein the timescales of extreme acceleration events are investigated in detail.

The $T_d$ from (B.1) needs to be perturbed to give a distribution of rise times, $P(T_d|\dot{s}_e)$, again to avoid that $T_d \dot{s}_e$ otherwise is
fixed at 6 m s$^{-1}$ for extreme $\dot{s}_e$. The deterministic expression (B.1) can be scaled stochastically by a factor following the dimensionless log-normal distribution

$$\frac{1}{w\sqrt{2\pi}} \cdot \exp\left\{-\frac{1}{2}\left[w^2 + \left(\frac{\ln x}{w}\right)^2\right]\right\}, \tag{B.2}$$

where the dimensionless width $w$ is small relative to 1; to begin we assume it is independent of $\dot{s}_e$. An example of the distributions $P(\dot{s}_e)$, $P(T_d)$, and $P(\dot{s}_e T_d)$ with $w = 0.2$, synthesized using $4 \times 10^6$ random values of $\dot{s}_e > 0.5$ m s$^{-2}$
following $P(\dot{s}_e)$ from the observed statistics at $z=100$ m and $f_c =1/3$ Hz, is shown in Figure 18. The actual width $w$, and its potential dependence on acceleration $\dot{s}_e$ (or wind speed), is the subject of future work.

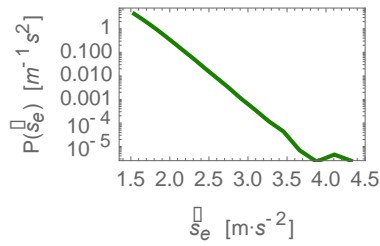 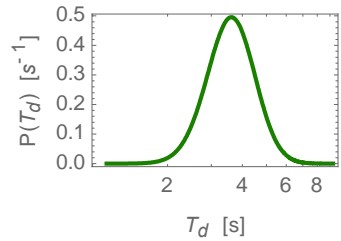 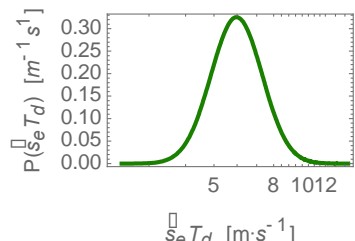

**Figure 18: example distributions of extreme acceleration (left), event duration (center), and associated wind speed jump (right) synthesized for simulation; $P(\dot{s}_e)$ used corresponds to $f_c = 1/3$ Hz at height of 100 m.**


Synthesized pairs of $\{\dot{s}_e, T_d\}$ can be used to drive a stochastic gust prescription (for e.g., aero-elastic simulations), given an analytical form for the gust. While the IEC 61400-1 standard has a deterministic gust, we propose a stochastic version which simply follows from the load-inducing increase in wind speed due to extreme flow accelerations and their statistics:

$$s(t) = U + \dot{s}_e \frac{T_d}{\pi}\left\{1 + \tanh\left[\frac{\pi(t-t_c)}{T_d}\right]\right\}, \tag{B.3}$$

where again the mean speed is $U$, and $t_c$ is the time corresponding to the peak acceleration (i.e., when $s = \dot{s}_e$). This has been formulated to give the simple acceleration waveform $\dot{s}(t) = \dot{s}_e \, \text{sech}[\pi(t - t_c)/T_d]$, and to allow convenient use of extreme $\dot{s}_{99}$ distributions (such as $P(\dot{s}_e)$ given above) and comparison with or extension of the standard EOG. The IEC standard prescribes $s_{\text{EOG}}(t) = U + 0.37 V_{\text{gust}} \sin(3\pi t/T_{\text{EOG}})[\cos(2\pi t/T_{\text{EOG}}) - 1]$ with a total duration of $T_{\text{EOG}} = 10.5$ s and





magnitude $V_{\text{gust}}$; it implies a maximum acceleration $\dot{s}_{e,\text{EOG}} = V_{\text{gust}}/(1.71\text{ s})$ which occurs at $t_c = 0.378T_{\text{EOG}} \simeq 4$ s. For

illustration and comparison, the extreme gust perturbations from the mean speed normalized by the maximum acceleration, $s'(t)/\dot{s}_e \equiv [s(t) - U]/\dot{s}_e$, are given in Figure 19 for the IEC's EOG along with the stochastic form (B.3). The latter is shown using three different values of rise time, $T_d = \{1.71, 4, 6\}$ s, corresponding to low, common, and high $T_d$ from the $P(T_d)$ shown in Figure 18; accompanying these normalized wind speed gusts are the corresponding $\dot{s}(t)$, also normalized by $\dot{s}_e$.

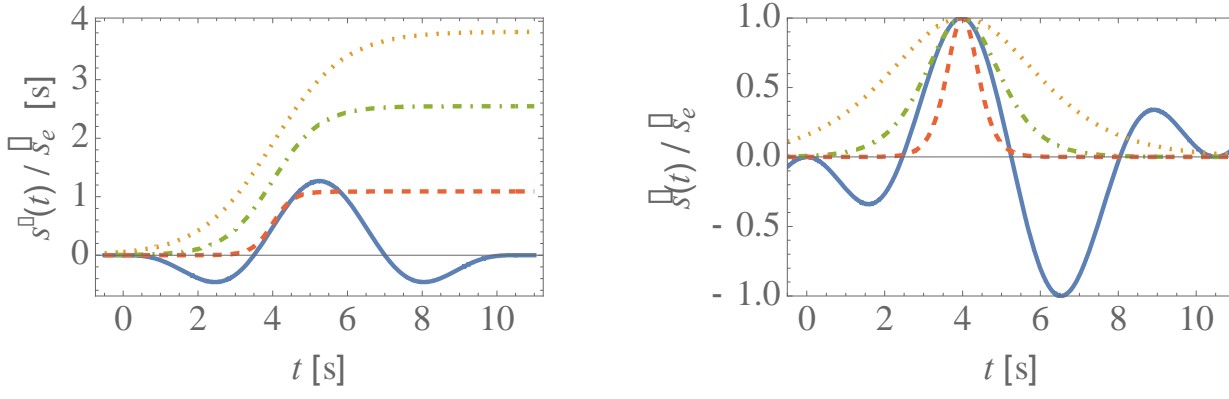


**Figure 19: wind speed gust function (left) and its time-derivative (right), normalized by peak acceleration EOG from IEC 61400-1.**

From the left-hand plot of Figure 19 we see that one can replicate the rising segment of $s_{\text{EOG}}(t)$ using (B.3) with $T_d = V_{gust}/\dot{s}_{\text{EOG}}(t_c) = 1.71$ s and consequently $\dot{s}_{e,\text{EOG}} = V_{\text{gust}}/(1.71\text{ s})$, or generically $\dot{s}_e T_d = \Delta s$; for longer rise times one then

sees larger wind speed 'jumps' for a given acceleration $\dot{s}_e$.

For multi-Megawatt turbine hub heights ($z_{\text{hub}} > 60$ m) and rotor diameters ($D \gtrsim z_{\text{hub}}$) we also see that the IEC's form for $V_{\text{gust}}$ is determined by its second term, expressible as $\dot{s}_{e,\text{EOG}} = \frac{I_{\text{ref}}(10.8\text{ m s}^{-2} + V_{\text{hub}}/0.69\text{ s})}{1 + 0.1D/42\text{ m}}$ where $I_{\text{ref}}$ corresponds to the IEC turbine intensity subclass while the hub-height wind speed is $V_{\text{hub}}$; the IEC 61400-1 implies EOG peak accelerations varying linearly from 2.2 to 3.6 m s$^{-2}$ for $I_{\text{ref}}$ of 12% (class C) and from 3.3 to 5.5 m s$^{-2}$ for $I_{\text{ref}}$ of 18% (class A+). These

*peak* EOG accelerations can exceed the 50-year amplitudes extrapolated from measurements by a factor of two for $f_c = 1/3$ s, and yet larger factors for lower $f_c$ (Figure 15); however, we see that the rising part (positive acceleration) of the EOG waveform, which lasts for 2.79 s (from $t = 2.46$ s to 5.25 s), has an average equal to $0.62\dot{s}_{e,\text{EOG}}$. For $I_u = 18\%$, and $V_{\text{hub}}$ ranging from 8–18 m s$^{-1}$, then $0.62\dot{s}_{e,\text{EOG}}$ ranges from 2–3.4 m s$^{-2}$; for $10 \lesssim V_{\text{hub}} \lesssim 16$ m s$^{-1}$ this falls between the 50-year values of $\dot{s}_e|_{f_c=1/3\text{ Hz}}$ statistically extrapolated from measurements at 100 m and 160 m height. Thus the IEC 61400-1 EOG

prescription and its inherent rise time of 2.79 s, for wind speeds near rated, implies an event-mean acceleration consistent with 50-year values extrapolated from measurements when considering the strongest turbulence subclass ('A+', 18%). For



lower IEC turbulence subclasses the EOG-implied event accelerations correspond to lower $f_c$; we point out that the IEC standard's prescription was originally developed onshore and closer to the ground, basically presuming gusts to be turbulent.

We remind (B.3) is meant to synthesize extreme acceleration timeseries for many different values of both $\dot{s}_e$ and $T_d$, as shown in the distributions of Figure 18; McWilliam, *et al.* (2023) did such synthesis, applying it through constrained turbulence simulations, for use in ultimate loads calculations.

*Competing interests*. The author declares he has no conflicts of interest.

*Acknowledgements*. The author would like to thank Michael McWilliam and Nikolay Dimitrov for fruitful discussions on usage of the methods; further, to Mike for discussions on and his use of the material in Appendix B, and to Nikolay for connecting to the ProbWind project and the IEC 61400-9 standard, and for supporting this work.

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
