# Peer review of "Flow acceleration statistics: a new paradigm for wind-driven loads, towards probabilistic turbine design"

_Wind Energy Science, 2024_

## Author Response (AR1)

**Final / summation of responses from author to reviews**

WES 2024-69

Below the author's point-by-point responses to the different reviews are given, in reverse chronological order.

**Response to Reviewer 2 (4.Jan.2025)**

This author thanks Reviewer 2 for their time and constructive comments (repeated below *in italic font)*. I reply to each separately, using the reviewer's respective comment numbers.

"*1) Research objective needs to be clarified in more clear way. Research gap and motivation were unclear because of the structure in abstract and introduction.*"

As mentioned in the abstract, motivations and objectives (which align with the development and results) include: calculation of a more robust flow statistic for loads, which allows accomodation for different turbines and controllers; solves the de-trending issue; permits (implicitly incorporates) description of significant fluctuations beyond just idealized turbulence, without making assumptions about spectra or PDF forms, accommodating atmospheric phenomena arising at heights far above the surface layer; it further facilitates identification of where IEC design-load case (e.g., EOG) prescriptions can be under-predicting (un-conservative, as cited/seen in application of this work). The motivation regarding the IEC 61400 standards is also re-stressed in more detail in the final paragraph of §2.0 (now lines 188–198).

"*2) For determination method of the wind acceleration in Section2, the author examined appropriate ways for the calculation by comparing the methods discretizing the wind speed numerically and based on power spectral density. However, Eq. (1) is not precisely correct because F[ds/dt] = (2πf)F[s], but not (2πf)²S$_{ss}$. Please explain why the inverse Fourier transform of the power spectral density corresponds to the wind speed acceleration.*"

The power spectral density (PSD) of wind speed ($S_{ss}$) is the Fourier transform of its autocovariance, thus the PSD of acceleration is $S_{\dot{s}\dot{s}} = \omega^2 S_{ss}$; eq.(1) was written in terms of PSD, since PSD of acceleration is plotted. However, your first statement is correct in the sense that the right-hand side of (1) is actually giving the autocovariance of acceleration. For the article and plots, I actually used the direct (simpler) calculation $\mathcal{F}^{-1}[-2i\pi f \mathcal{F}[s(t)]]$ but wished to avoid complications around programming language-dependent aspects of inverse FFT's, i.e., to "keep it simple" and streamlined.

I have now corrected (1), also to show what is directly calculated, and added a footnote in the text following it for caution, clarity, and repeatability.

"*3) In Section 2.1 and Figure 2, the authors showed the effect of the filter on the max and top percentiles; however, the decrease of the values by increasing the cut-off filter time scale is very natural consequences. This reviewer could not follow the physical meaning to show these results as the preliminary demonstration.*"

As written, increasing the filter timescale (reducing $f_c$) reduces the acceleration level for a given exceedance probability. The 'physical' meaning is that lower $f_c$ means filtering out more of the fluctuations, i.e. at timescales smaller than $1/f_c$. I add a sentence along these lines now just before Figure 2. Further, Appendix B includes information about this.

"*4) As for Figure 4, the top 1-10 values are explained, but they are unclear in Figure 4. From the cumulative density in the vertical axis, it is unclear which values correspond to certain percentiles.*"

The minimum SF is noted, so the top 10 values follow as 1-10 times that; Figure 4 would be too busy if every point was plotted separately, given its logarithmic scale.

"*5) Figure 6 and 7 are very interesting to see the relationship between the acceleration and statistics. The results are clearly presented to clarify the linear relationship between the acceleration and standard deviation. This reviewer recommends adding the physical.*"

It is not clear what is meant by the recommendation of "adding the physical". Some interpretation is given for the speed dependence in the text, for the most common values of $\dot{s}_{99}|S$ and $\dot{s}_{99}|\sigma_s$. However, physical interpretation of the largest $\dot{s}_{99}$ is not straightforward, as large $\dot{s}_{99}$ arise due to a number of different mechanisms; this is noted and discussed in the sections afterward.

**Response to Reviewer 1 (4.Oct.2024)**

Thank you for the comments. I'll reply to them individually.

"*1. Clearer and higher resolution images are more important for reader understanding.*" These will be provided in the final PDF document.

"*2. Can you consider adding more model evaluation indicators?*" It is not clear what model you refer to. The main contribution/advancement of this work is the conception of filtered flow-acceleration metrics towards statistical characterization of transients for better loads simulation; there is not really a predictive model here to evaluate. However, estimation of the 10-minute P99 of filtered acceleration expected for a given return period is given; as a conservative indication of (maximum) uncertainty in this, Fig.15 includes bands showing the possible variation in the result due to the full range of possible base periods.

"*3. Show some comparison results with recent studies.*"

This request is a bit unclear, as the work describes a new paradigm (and methodology) — exceedence statistics of filtered flow accelerations — whose 'results' are given statistically. The paper refers to works by McWilliam *et al.*, who have used it for loads calculations and *comparisons* with e.g. the IEC 61400-1 standard. Further comparisons are beyond the scope of this (already quite long) article, and left for follow-up investigations.

**Response to Associate Editor's report**

**To** Etienne Cheynet

**Reg.** associate editor report for WES-2024-69

**From** Mark Kelly                                                                 20 June 2024

**Response to Assoc. Editor report for WES-2024-69**

Hi Etienne,

Thanks for the constructive comments. I paste your report in *italicized form*, responding pointwise using blue font below. Given your interpretations and ensuing responses/discussion, it would have been nice to include this in the WES article discussion. Will this/the below (hopefully) be publicly available?

*This manuscript is within the scope of the journal and appears to meet its basic scientific quality. I find this draft very interesting and agree with the idea of exploring flow acceleration alongside traditional wind velocity measures.*
*However, I have a few comments that the author may consider:*
*The draft chooses to focus on the horizontal wind speed component 's'; however, it might be prudent to warn the reader that using 's' for wind load design is not recommended. More specifically, the standard deviation of the along-wind component is used in IEC 61400-1 (2019), see eqs. 10, not directly the standard deviation of the wind speed. Please do not hesitate to notify me if I am wrong. From my understanding, the standard deviation of wind speed, i.e., the magnitude of the velocity components, should not be used for wind load calculation as it results in an underestimation of the along-wind loading (sigma_u > sigma_v > sigma_w, where u v and w are the three velocity components as by Kaimal and Finnigan (1994)).*

I agree to better "warn" the reader, since most of them might presume it is 'simply ABL turbulence' (especially if they only skim without reading Sect.4 or §3). This includes clarification to avoid potential misunderstandings, related to your response regarding the IEC 61400-1: in industrial practice, what is presumed to be the longitudinal standard deviation ($\sigma_u$), is not typically measured as such. To the contrary, from the IEC 61400-50-1, 61400-12-2, and 61400-12-1 standards, measurement of the standard deviation of horizontal wind speed ($\sigma_s$) is prescribed when using cup anemometers. It is due to this standard practice that I consider $\dot{s}$ and $\sigma_s$, to be responsibly explicit and also promote awareness of this aspect. Kristensen's 2000 paper (eq.24 with eq.12) also shows that cup-anemometers give $\sigma_s$, though one has to dig into the equations a bit; Yahaya & Frangi (2004) also write that cup anemometers give $\sigma_s$.

I further point out that $\sigma_s > \sigma_u$ (see e.g. Kristensen 1998 & 2000), which leads to over-estimation (possibly why the practice of using $\sigma_s$ persists, as a conservative choice), not an under-estimation in streamwise loading.
But the practice of using $\sigma_s$ from cup-anemometers and treating it as $\sigma_u$ (i.e., without using a windvane to obtain high-frequency $u(t)$ to get $\sigma_u$, despite Kristensen's (2000) recommendation) is beyond the scope of this article, and is the subject of parallel ongoing work which includes the conditional behavior of $\sigma_u/\sigma_s$.
As mentioned in the manuscript, for extreme flow accelerations — i.e., the largest positive magnitudes from many years of 10-minute $P_{99}(\dot{u})$ and $P_{99}(\dot{s})$ values — we see $P_{99}(\dot{s}) = P_{99}(\dot{u})$. Prior to submission I originally made Fig.14 to include traces of $u(t)$ along with $s(t)$ to demonstrate this; I include such plots as an Appendix below. But since there is such full overlap of $u$ and $s$ over the gusts/'jumps' (deviating only rarely during a small part of each 10-minutes away from the $\dot{s}_{99}$ 'jump' segment), and because the extreme directional parts are not a focus

(as explained in the manuscript), I chose to not to include the $u(t)$ in Figs.14-15, in order to avoid distracting or confusing the reader.

I also didn't show the $u(t)$ or get further into this, for two more reasons:

1. using an anemometer with separate vane is standard for wind-industry measurements, but such setups can degrade/ruin single-point statistics for $\dot{u}$ (and to some extent $u$) due to the instrument separation distance and consequent directional dependence/lag. On our mast they were separated by <1 m, crucially in the (roughly) transverse direction for the offshore flow considered.
2. It requires much more work — and another article — to investigate and characterize how $\dot{s} \simeq \dot{u}$ (or possibly $s \approx u$) for the strongest fluctuations, especially given the number of physical phenomena (some of them non-turbulent) causing the extremes, along with the associated height-dependence of the strength and frequency of different phenomena. I discuss the latter aspects in Sect.4 already. I also note/remind here that wind tunnel studies are not appropriate for this, and theoretical fluid dynamics has not yet advanced sufficiently into this area (atmospheric flow affected by the chaotic capping inversion) which goes beyond turbulence and still requires significant research.

I note that $P_{99}(\dot{s}) \simeq P_{99}(\dot{u})$ to a good extent for non-extremes also. ABL turbulence theory (e.g., K&F1994; Wyngaard, 2010) does not cover the PDF's "tails" (i.e., here the top 1%) well, where fluctuations become less Gaussian — and where aberrations from classical/mean theory are yet greater for accelerations than for velocities.

Again this work is not focused on turbulence per se, due to the phenomena which cause the largest flow accelerations; I emphasized this in the discussion.

I add more 'warning'/explanation along these lines into the manuscript, to help.

*I suggest being cautious with the high rate of self-citation (over 30% is unusually high), as it may indicate limited engagement with the existing literature. For example, there has been a large body of less-known work on non-stationary wind models from the wind engineering community that may have been overlooked by the author.*

I agree with the general guideline, but in this case, it is not due to limited engagement. There are now 41 citations which do not include me, indicating engagement with literature that spans multiple disciplines; adding several more (e.g., to explain use of $\{\dot{s}, \sigma_s\}$, use/implications of cup anemometry), the self-citation rate is now 25%. The relatively high rate arises for several reasons: very few others have looked at atmospheric accelerations at this scale; I have been very active in the meteorological (and probabilistic ABL) aspects needed to explain/interpret findings here; and this work builds on the combination of work I've done in various sub-areas, which is why it happened (the combined expertise to conceive and do the work). I also attempt to cite parallel contemporaneous work done by others, not just works involving myself; I believe this is good practice, supporting the scientific method.

*Indeed, non-stationary wind models include non-zero flow acceleration, although this is not always explicitly stated. I believe that previous studies on non-stationary flow modelling may complement the present draft. A key reference may be Kareem, A., Hu, L., Guo, Y., & Kwon, D. K. (2019). Generalized wind loading chain: Time-frequency modeling framework for nonstationary wind effects on structures. Journal of Structural Engineering, 145(10), 04019092. I enclose a copy of this paper.*

This work is not about non-stationary modelling, as discussed in the subsections describing the spatial high-pass filtering; this author thanks you for those references, but they do not appear to be relevant here. (I do add however that I have been looking into effective ergodicity spectrally via scale separation in other ongoing research involving conditional filtering in 1d & 2d, and note for [non]stationarity in the inhomogeneous ABL one can also look at Pan & Patton, 2017.)

*The term 'new paradigm' may be controversial, as wind engineering has explored concepts of non-*

*stationary turbulence and probabilistic loads since the early 2000s, particularly for long-span bridges and towers. I agree with the author that a new paradigm may emerge from the current draft, but maybe such a shift has already started in contemporary works.*

Again, this work is not about non-stationarity and not focused on turbulence. Further, confirming with dozens of colleagues/experts about this since ~2021 (while active in fluid dynamics/mechanics, meteorology, signal processing, wind engineering), I stand by this provocative label: filtered flow accelerations have not been considered in wind energy (and additionally accommodating turbine response); and because this paradigm goes beyond turbulence as well as the standard 10-minute averages used in wind energy. Further, as explained in the introduction, where flow accelerations had been considered before for wind, they were dismissed due to their perceived intractability since low-pass filtering had not been considered; note Sect.1 I already includes Nielsen *et al.*'s (2004) mention of the potential need for low-pass filtering in dealing with gusts (as well as how it was conceptually different).

   A 'shift' towards probabilistic has indeed happened in contemporary works (I cite some of this, including some of my own), but the current manuscript also involves filtering along with a novel meta-probabilistic aspect — using $P_{99}$ of accelerations within 10-minute periods then building long-term statistics of that — which demands something beyond currently used 10-minute statistics.  The novelty is also reflected in the IEC 61400-9 having interest in incorporating such. So currently I stand by the 'new paradigm' phrase, unless more review(s) argues solidly to modify/remove it.

*Line 80: "We note that although a couple of studies aimed at statistics of gust-like events have recently appeared in the literature, they did not focus on offshore load-inducing flow characterization." I partly agree here, but we should remember that gust-like events have motivated the development of wind engineering since the 1960s. Gust-like events were studied through the concept of peak loading for example. Offshore gust-like events were used to design offshore platforms for oil and gas operations in the 1980s. This concept of "peak loading", although (over-)simplistic, is nowadays used in standard and code, e.g. Eurocode or ESDU standard.*

Thank you for the reminder; the point was that gust-like events beyond the surface layer (or affected by the capping inversion) offshore have not been characterized much, as offshore bridges are well within the surface layer.
I add more text relating this difference and mentioning/citing the historical civil engineering aspect.

*If I recall correctly (per the Wiener–Khinchin theorem), the right-hand side of Equation 1 should use the square of the circular frequency (omega = 2*pi*f) rather than the square of the frequency f. Otherwise there may be a scaling issue. This can be checked using acceleration and displacement data from a simple vibration system.*

Thanks for catching the typographical error. I had mixed notation (originally wrote $\omega^2 S_{uu}(f)$) and when changing to be only in terms of $f$ I forgot the $2\pi$—but in the calculation it is $(2\pi f)^2 S_{uu}$. If you look at Figure 1 and the text around it, you can see that the correct form has been used; e.g., the figure shows spectra calculated both using finite-difference and spectrally with $(2\pi f)^2 S(f)$. Eq.1 is now corrected.

*Line 336-337: "Previous works have presumed that extreme load-driving flow phenomena tend to be associated with non-Gaussian turbulence," is, in my understanding, due to the non-stationary nature of extreme loading but once the non-stationary and stationary fluctuations are separated, the stationary part may be Gaussian. However, I may be wrong here. See e.g., Chen, J., Hui, M.*

*C., & Xu, Y. L. (2007). A comparative study of stationary and non-stationary wind models using field measurements. Boundary-layer meteorology, 122, 105-121.*

This may be true for (most) turbulence, but I remind that the dominant timescales of the flow accelerations are shorter than 1 minute (or even <10 s for extremes with $f_c$ =1/3 Hz), and that the extremes are mostly not due to turbulence. Non-stationarity involves much longer timescales; while one can remove some non-stationarity through high-pass filtering (which is also done and considered here), the strongest accelerations remain and are not readily 'separated'.
Thank you for the Chen *et al.* (2007) reference, which I add in l.338 for context; I also add text (just above Fig.8) to more clearly explain that extreme accelerations are found without non-stationarity, connecting with Sect.3.3.

*The y-labels in Figures 14 and 15 are a little challenging to read. Maybe the design of the figure can be optimized by showing only the x-label for the last panel?*

The plots have now been re-made for easier reading of the labels.
I have also removed Figure 15, because it doesn't give any additional information than Fig.14 other than different events/dates are detected when using $f_c$=1/3 Hz instead of 0.1Hz; this was noted in the text.
(Further, there are already many figures, the paper is long, and Figs.14 & 15 were the largest.)

*(Gaussian) wind loading is driven by mainly three elements: (1) the mean wind speed; (2) the 1-point velocity spectra and (3) the spatial-correlation of gusts (coherence). The draft focuses on the first two points. Maybe a few words for the last point can be interesting for the reader.*

Regarding spatial correlation, I discussed the streamwise and transverse length scales of the flow structures responsible for the observed extreme accelerations, in Sect.4; for example, the need for a sufficiently large transverse length scale to appreciably impact turbine loads. I did not include spectral coherence because most observed extreme $\dot{s}_{99}$ (gusts) had durations shorter than 30s, so $\text{Coh}(f, \Delta z)$ very noisy due to the short sampling and only 2 measurement points; further, sometimes there was also a lag between 100m and 160m heights. To be direct, understandable, and avoid speculation, I thus examined $d\dot{s}/dz$ in Sect. 3.1.1 (and Fig.9); in Sect.4 I also concluded about the implied scales, along with the need to measure the scales and extreme flow structures more directly in Sect.4.1. Now per the latter, I add multi-point measurements to this.

**References**

International Electrotechnical Commission: *IEC 61400-1, Wind turbines – Part 1: Design requirements, 4th ed.*, International Electrotechnical Commission, Geneva, Switzerland, ISBN 978-2-8322-6253-5, 2019.

International Electrotechnical Commission: *IEC 61400-12-1, Wind energy generation systems – Part 12-1: Power performance measurements of electricity producing wind turbines, 3rd ed.*, International Electrotechnical Commission, Geneva, Switzerland, ISBN 978-2-8322-5621-3, 2022.

International Electrotechnical Commission: *IEC 61400-12-2, Wind energy generation systems – Part 12-2: Power performance of electricity producing wind turbines based on nacelle anemometry, 2nd ed.*, International Electrotechnical Commission, Geneva, Switzerland, ISBN 978-2-8322-5594-0, 2022.

International Electrotechnical Commission: *IEC 61400-50-1, Wind energy generation systems – Part 50-1: Wind measurement – Application of meteorological mast, nacelle and spinner mounted instruments*, International Electrotechnical Commission, Geneva, Switzerland, ISBN 978-2-8322-5937-5, 2022.

International Electrotechnical Commission: *Technical Specification 61400-9, Wind energy generation systems – Part 9: Probabilistic design measures for wind turbines, 1st ed.*, International Electrotechnical Commission TC88, Geneva, Switzerland, 2023.

Kristensen, L.: "Cup Anemometer Behavior in Turbulent Environments", *J. Atmos. Oceanic Tech.* **15** (1), 5–17, https://doi.org/10.1175/1520-0426(1998)015%3C0005:CABITE%3E2.0.CO;2, 1998.

Kristensen, L.: "Measuring Higher-Order Moments with a Cup Anemometer." *J. Atmos. Oceanic Tech.* **17** (8): 1139–48, https://doi.org/10.1175/1520-0426(2000)017<1139:MHOMWA>2.0.CO;2, 2000.

Yahaya, S., & Frangi, J. P.: "Cup anemometer response to the wind turbulence — Measurement of the horizontal wind variance", *Ann. Geophys.* **22**(10), 3363–3374. https://doi.org/10.5194/angeo-22-3363-2004, 2004

**Appendix**

Below (Figs. AC1–AC2) we show the 10-minute records for the strongest acceleration events in each speed bin for $f_c$=1/3 Hz, at 100m and 160m heights respectively. Other than ~1.5 minutes of the 12m/s bin case at 160m, one sees that $s(t) = u(t)$ in the plots.

[Figure]

**Figure AC1: strongest $\dot{s}$ events at $z$=100m, for $f_c$=0.1s.** Blue: $s|_{z=100m}$; gold: $s|_{z=160m}$; green: $u|_{z=100m}$; red: $u|_{z=160m}$.

[Figure]

**Figure AC2: strongest $\dot{s}$ events at 160m, for $f_c$=0.1Hz. Blue: $s|_{z=100m}$; gold: $s|_{z=160m}$; green: $u|_{z=100m}$; red: $u|_{z=160m}$.**